# Defining the biogeographical map and potential bacterial translocation of microbiome in human 'surface organs'

Jun-Jun She[1,2,3,4,8] ✉, Wei-Xin Liu[5,8], Xiao-Ming Ding[6,7,8], Gang Guo[2,3,8], Jing Han[2,3,8], Fei-Yu Shi[1,2,3], Harry Cheuk-Hay Lau[5], Chen-Guang Ding[6,7], Wu-Jun Xue[6,7], Wen Shi[2,3], Gai-Xia Liu[1,2,3], Zhe Zhang[1,2,3], Chen-Hao Hu[1,2,3], Yinnan Chen[2,3], Chi Chun Wong[5] & Jun Yu[2,5] ✉

The microbiome in a specific human organ has been well-studied, but few reports have investigated the multi-organ microbiome as a whole. Here, we aim to analyse the intra-individual inter-organ and intra-organ microbiome in deceased humans. We collected 1608 samples from 53 sites of 7 surface organs (oral cavity, esophagus, stomach, small intestine, appendix, large intestine and skin; $n = 33$ subjects) and performed microbiome profiling, including 16S full-length sequencing. Microbial diversity varied dramatically among organs, and core microbial species co-existed in different intra-individual organs. We deciphered microbial changes across distinct intra-organ sites, and identified signature microbes, their functional traits, and interactions specific to each site. We revealed significant microbial heterogeneity between paired mucosa-lumen samples of stomach, small intestine, and large intestine. Finally, we established the landscape of inter-organ relationships of microbes along the digestive tract. Therefore, we generate a catalogue of bacterial composition, diversity, interaction, functional traits, and bacterial translocation in human at inter-organ and intra-organ levels.

Surface organs (organs with direct or indirect contact with the ambient air), including the oral cavity, gastrointestinal tract (GI), and skin, are the largest organs in human body with widespread involvement in physiological, metabolic, and immunological processes. Surface organs are inhabited by trillions of microbes and harbored diverse microbial communities[1,2]. The location and functional features of each surface organ create specific environmental conditions that gives rise to regional specificity in microbial populations[3]. To date, emerging studies focusing on regional differences have unveiled numerous physiological roles of the microbiome that are crucial for human life[4,5]. Whereas few has reported the heterogeneity in microbiome among different body parts[4]. Such inter-organ microbial disparity also contributes to variation and plasticity in the functional traits of gut microbiome[6]. However, only a few investigations deciphered the microbiome along the digestive organs with limited sampling sites from the same individual[7]. To fully uncover the human microbiome, it is necessary to assess microbiome composition in

[1]Department of General Surgery, First Affiliated Hospital of Xi'an Jiao Tong University, Xi'an, China. [2]Center for Gut Microbiome Research, Med-X Institute Centre, First Affiliated Hospital of Xi'an Jiao Tong University, Xian, China. [3]Department of Talent Highland, First Affiliated Hospital of Xi'an Jiao Tong University, Xi'an, China. [4]Yulin Hospital, First Affiliated Hospital of Xi'an Jiao Tong University, Yulin, China. [5]Institute of Digestive Disease and The Department of Medicine and Therapeutics, State Key Laboratory of Digestive Disease, Li Ka Shing Institute of Health Sciences, CUHK Shenzhen Research Institute, The Chinese University of Hong Kong, Hong Kong SAR, China. [6]Department of Kidney Transplantation, The First Affiliated Hospital of Xi'an Jiao Tong University, Xi'an, China. [7]Institute of Organ Transplantation, Xi'an Jiao Tong University, Xi'an, China. [8]These authors contributed equally: Jun-Jun She, Wei-Xin Liu, Xiao-Ming Ding, Gang Guo, Jing Han. ✉e-mail: junjunshe1975@sina.com; junyu@cuhk.edu.hk

surface organs of digestive system (lumen and mucosa) and skin with much denser sampling as a whole.

Microbiome in different regions/or sites of an organ could be varied, exemplified by the difference in microbiome between proximal and distal colon[3]. We recently reported the presence of microbial heterogeneity in a single colorectal tumor[8]. Understanding whether there are discernible patterns of microbial communities along the GI tract and elucidating regional microbial niches, are of great importance for complete mapping of the human microbiome. However, it has been challenging to collect samples from multiple sites within an organ, especially in the small intestine. Moreover, it remains undetermined whether microbiome in lumen samples is similar or different from microbiome in mucosa samples of the whole GI tract.

The microbial crosstalk among organs is emerging as an essential indicator of human health[9,10]. Yet, little is known about the routine inter-organ contacts of the microbiome. For instance, how the oral microbial community correlates with microbiome in other organs is completely unknown. Sampling a breath of intra-individual surface regions therefore offer an opportunity to decipher how microbial residents might translocate from one site/organ to another, respond to changing environments, and shape host physiology.

In this study, lumen mucosa, gastric juice, and surface samples from 53 sites of 7 surface organs (oral cavity, stomach, esophagus, small intestine, appendix, large intestine, and skin) were collected from 33 subjects to give a total of 1608 samples. The large collection of samples facilitated the generation of a high-resolution biogeographical map of the human microbiome. Our findings revealed the differences in diversity, composition, interaction, and functional traits of microbiome among organs in the digestive system and all surface organs, as well as those among different sites within an organ. We also

profiled the luminal- and mucosal-associated microbes along the GI tract. Using 16S full-length data from PacBio highly accurate long-read sequencing, we finally elucidated the inter-organ relations of microbes at species level in human body.

## Results

### Sample collection for microbiome profiling

We collected 1608 samples from 7 surface organs of oral cavity (6 sites), esophagus (4 sites), stomach (5 sites), small intestine (14 sites), appendix (1 site), large intestine (13 sites), and skin (10 sites), in total comprising of 53 sites in 33 subjects (Fig. 1A) who were dead due to vehicle accident, high-altitude falling, etc. (Supplementary Table 1). To minimize the post-mortem microbial changes, all samples were collected in a short duration (<1.5 h) after determination of death. Both luminal and mucosal samples were collected from the stomach, small intestine and large intestine. We parallelly introduced a set of negative controls to evaluate potential contamination (Supplementary Fig. 1A). All retrieved samples were subjected to microbial profiling by 16S v3v4 region sequencing, and samples from GI organs ($n = 1030$) were additionally analyzed by PacBio 16S full-length HiFi sequencing. After eliminating ASVs detected in negative controls (Supplementary Fig. 1B and Supplementary Table 2), we obtained a total of 9473 bacterial ASVs for downstream analysis (Fig. 1B). Key contaminating ASVs consist of environmental taxa (e.g., *Propionibacterium* (17.08%; relative abundance in negative controls), *Phyllobacterium* (6.12%), *Deinococcus* (4.87%)), and they were on average one order of magnitude higher than in mucosal samples as compared to luminal samples (Supplementary Table 2). We next applied permutational multivariate analysis of variance (PERMANOVA) to study the effect of subject's characteristics (e.g., cause of death, length of hospitalization) on the

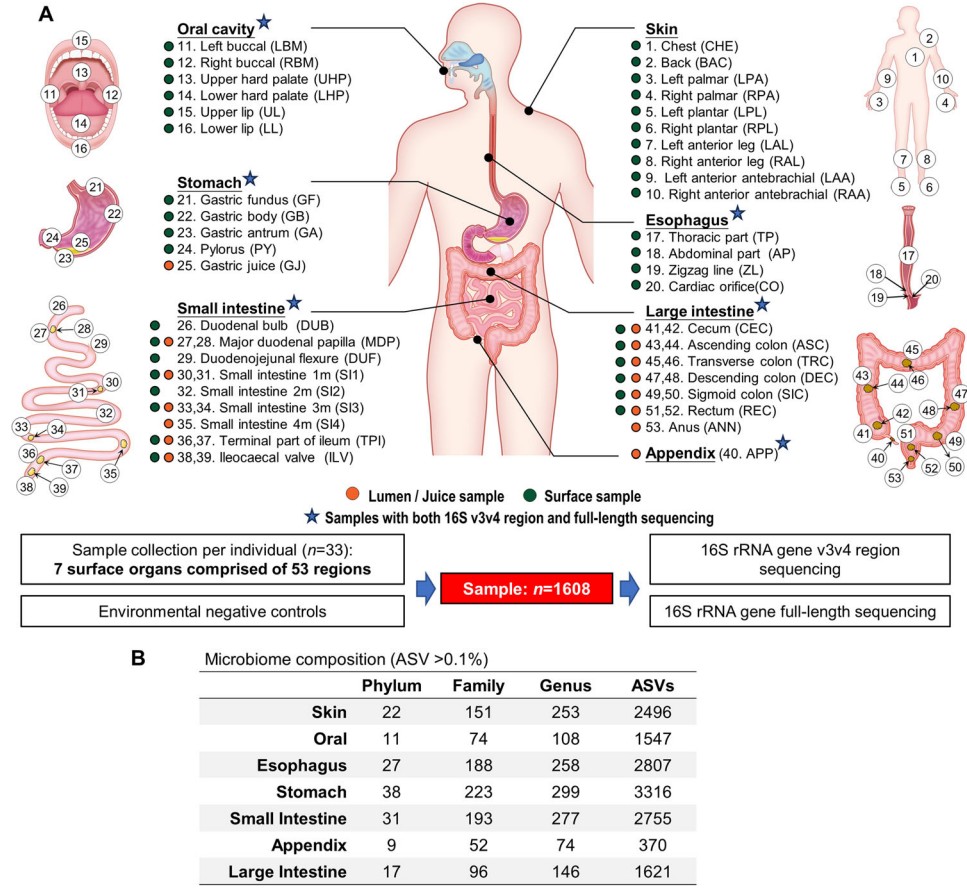

**Fig. 1 | Body-wide microbiome profiling in human subjects. A** A total of 1608 samples from 53 body sites of 7 surface organs were collected from 33 subjects and were subjected to microbiome profiling. **B** The amount of detectable phylotypes in each organ at different taxonomic levels.

Microbiome composition (ASV >0.1%)

| | Phylum | Family | Genus | ASVs |
|---|---|---|---|---|
| **Skin** | 22 | 151 | 253 | 2496 |
| **Oral** | 11 | 74 | 108 | 1547 |
| **Esophagus** | 27 | 188 | 258 | 2807 |
| **Stomach** | 38 | 223 | 299 | 3316 |
| **Small Intestine** | 31 | 193 | 277 | 2755 |
| **Appendix** | 9 | 52 | 74 | 370 |
| **Large Intestine** | 17 | 96 | 146 | 1621 |

microbiome communities. We found the length of hospitalization and antibiotic treatments had significant effects on microbiome in the oral cavity, small intestine, and large intestine, but not the cause of death (Supplementary Table 3).

## Microbial diversity varies among surface organs

We first investigated the bacterial diversity in surface organs. Significant differences in bacterial α-diversity were identified among surface organs (Fig. 2A and Supplementary Fig. 1C). The α-diversity of skin, oral cavity, and esophagus was significantly higher compared to stomach, appendix, small or large intestines, respectively ($P < 0.01$, Wilcoxon signed-rank test). Among seven organs, the bacteria diversity in stomach was the lowest, attributed to its low pH that limits bacterial growth. Significantly higher α-diversity in the large intestine was observed when compared to stomach or small intestine ($P < 0.05$).

Changes in α-diversity along the GI tract were then measured. In the upper GI tract (esophagus-stomach-duodenum), we observed that α-diversity initially falls in esophagus, reaching the bottom at the stomach, and subsequently rising at the duodenum ($P < 0.05$) (Fig. 2B). Meanwhile, significantly increasing trend of α-diversity was also identified in luminal samples along the lower GI tract (jejunum-Ileum-colon) ($P < 0.05$) (Fig. 2B), mainly attributed to the longer transit time in colon. When comparing α-diversity between mucosal and luminal samples, we observed disparities in α-diversity along the lower GI tract (Fig. 2B). Specifically, mucosal α-diversity was higher in jejunum/Ileum ($P < 0.05$) compared to luminal samples; while mucosal α-diversity was lower than that of luminal in the large intestine ($P < 0.0001$).

The global microbial β-diversity was also significantly different among organs ($P < 0.001$, PERMANOVA) (Fig. 2C). The most different microbiome was found between the large intestine and oral cavity,

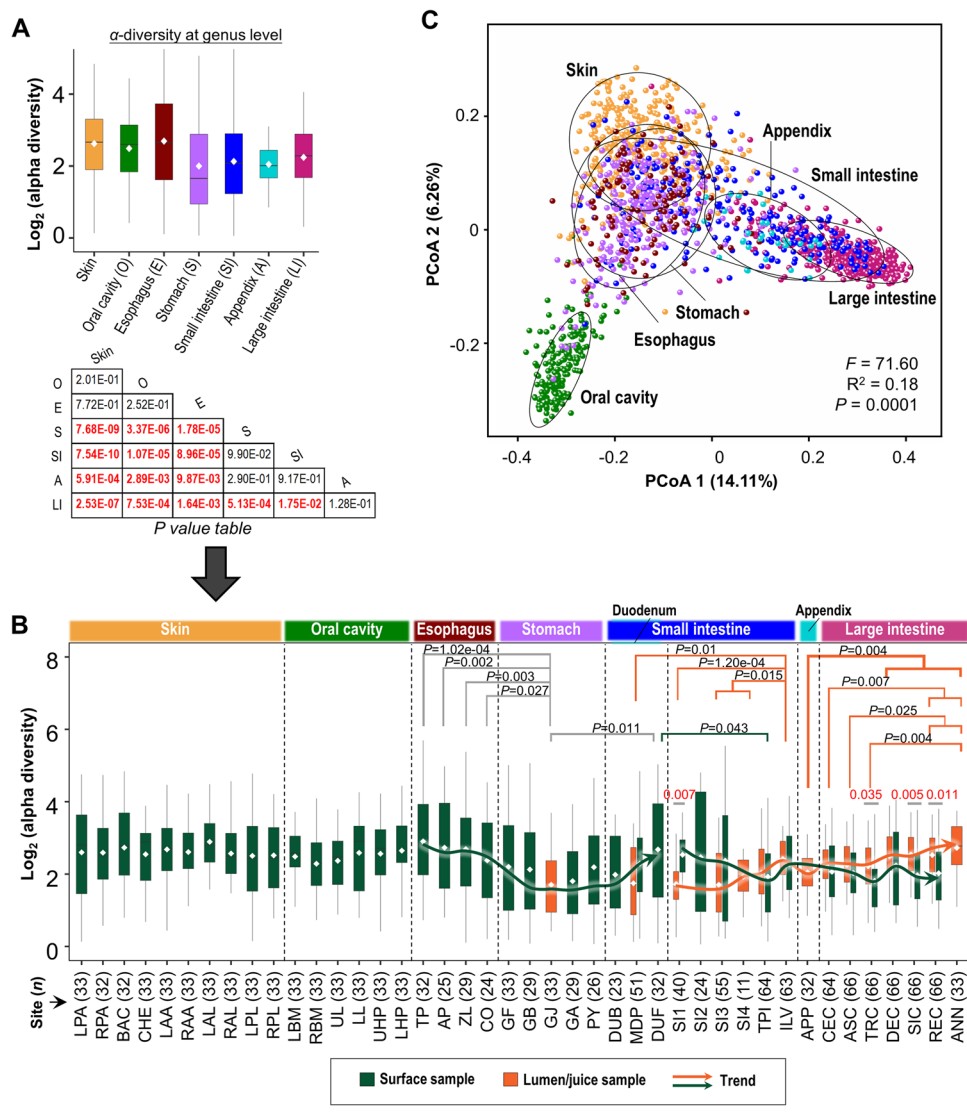

**Fig. 2 | Microbial diversity among seven organs. A** α-diversity of samples was grouped by organs ($n = 328, 198, 110, 150, 363, 32, 427$ for skin, oral cavity, esophagus, stomach, small intestine, appendix, and large intestine, respectively) and measured using the relative inverse Simpson index at the genus level. Boxplots were colored by surface organs. *P* values were determined using two-sided Wilcoxon signed-rank test. **B** α-diversity of 53 body sites in surface organs (sample size *n* was indicated in the button of each boxplot). Boxplots and trendlines were colored by sample types (surface or lumen). *P* values were determined using two-sided Wilcoxon signed-rank test. **C** β-diversity was measured using PCoA based on UniFrac distance. Each point (sample) was colored by its belonged organ. Community dissimilarities were tested by PERMANOVA analysis. Data are shown as Box and whisker plots (**A, B**) to represent the median (center line), quartiles (box), and range (whiskers) of the α-diversity for each community, excluding outliers (points outside 1.5 times the interquartile range). Source data are provided as a Source Data file.

whilst microbiome between the stomach and esophagus was the least different (Supplementary Table 4). In the small intestine, we observed drastic intra-organ variation (Fig. 2C), and showed that intra-organ variation in the small intestine spans between stomach and large intestine clusters according to the sampling location (Supplementary Fig. 1D). Additionally, appendix microbiome was similar to the small intestine microbiome (Fig. 2C and Supplementary Table 4).

### Inter-organ microbial communities are distinct

We next measured the inter-organ microbial composition. Six phyla (*Proteobacteria, Firmicutes, Bacteroidetes, Actinobacteria, Fusobacteria,* and *Tenericutes*) together occupied >98% relative abundance in each organ (Fig. 3A1 and Supplementary Fig. 2A). Abundances of these phyla were all significantly different among seven organs (Fig. 3A2). *Bacteroidetes, Actinobacteria,* and *Fusobacteria* were enriched in large intestine, skin, and oral cavity, respectively, while *Proteobacteria* and *Firmicutes* were enriched in esophagus, stomach, and small intestine. Microbial composition at genus level was then assessed (Fig. 3B). *Bacteroides* and *Parabacteroides* were predominantly enriched in small intestine, appendix, and large intestine; *Porphyromonas, Prevotella, Streptococcus* and *Neisseria* were enriched in the oral cavity; *Fusobacterium* was enriched in both oral cavity and appendix; and *Staphylococcus,* and *Corynebacterium* were the dominant genera in the skin. At the individual level, we observed a decreasing trend in the abundances of *Staphylococcus* and *Corynebacterium* from the skin to

GI tract (Supplementary Fig. 2B). Conversely, increased abundances of *Enterococcus, Ruminococcus* and *Bifidobacterium* were observed along the GI tract. Moreover, *Helicobacter* was enriched in stomach and esophagus. These findings together suggested that microbial composition differs among surface organs.

### Intra-organ microbial communities are heterogenous

As shown in Supplementary Fig. 2, microbes were not evenly distributed in each organ. We therefore investigated the microbiome of different intra-organ sites. β-diversity was significantly different among sites of each organ (Fig. 4). We identified the signature microbes specific to each site in an organ: *Corynebacterium* and *Staphylococcus* in the extremity cluster in skin (Fig. 4A); and *Aggregatibacter* in the jaws cluster of the oral cavity (Fig. 4B). In the esophagus, *Helicobacter* was increased from Thoracic Part (*TP*) to Cardiac orifice (*CO*), while its abundance was decreased from Fundus/Body to Pylorus (*PY*) in the stomach (Fig. 4C, D). In the small intestine, *Prevotella* were enriched in both the mucosa and lumen of duodenum, whereas *Enterococcus* and *Bacteroides* were enriched in both the mucosa and lumen of ileum (Fig. 4E, F). In the large intestine, we identified clear separation of microbiome between the right-sided and left-sided colon, attributed to the disparity in the enriched microbes (*e.g., Klebsiella* in the right-sided colon; *Bifidobacterium* and *Oscillospira* in the left-sided colon) (Fig. 4G, H). These data revealed the distinct microbial composition of intra-organ sites.

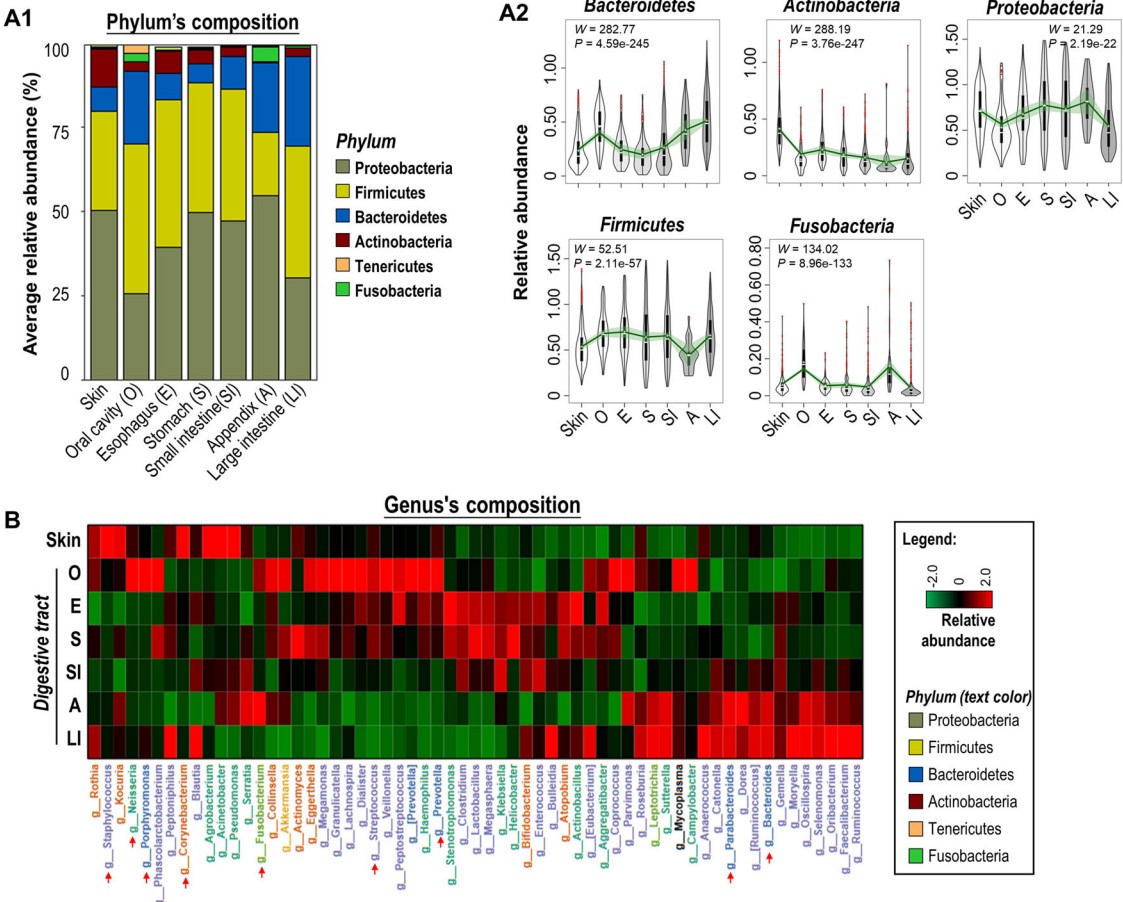

**Fig. 3 | Microbiome composition among seven surface organs. A1** Abundances of six major phyla in seven organs: *Proteobacteria* (relative abundance: 41.31% ± 9.63%, Mean ± SD), *Firmicutes* (35.02% ± 8.36%), *Bacteroidetes* (14.10% ± 8.30%), *Actinobacteria* (6.21% ± 5.46%), *Fusobacteria* (1.65% ± 1.80%), and *Tenericutes* (0.37% ± 0.83%). **A2** Phylum with significantly different abundance among seven organs by ANCOM-BC2 method (*n* = 328, 198, 110, 150, 363, 32, 427 for skin, oral cavity, esophagus, stomach, small intestine, appendix, and large intestine, respectively. **B** Genus with significantly different abundance among seven organs by ANCOM-BC2 method. The colormaps represents the average bacterial abundance. Data are shown as Box and whisker plots (**A2**) to represent the median (center line), quartiles (box), range (whiskers), and outliers (points outside 1.5 times the interquartile range). Source data are provided as a Source Data file.

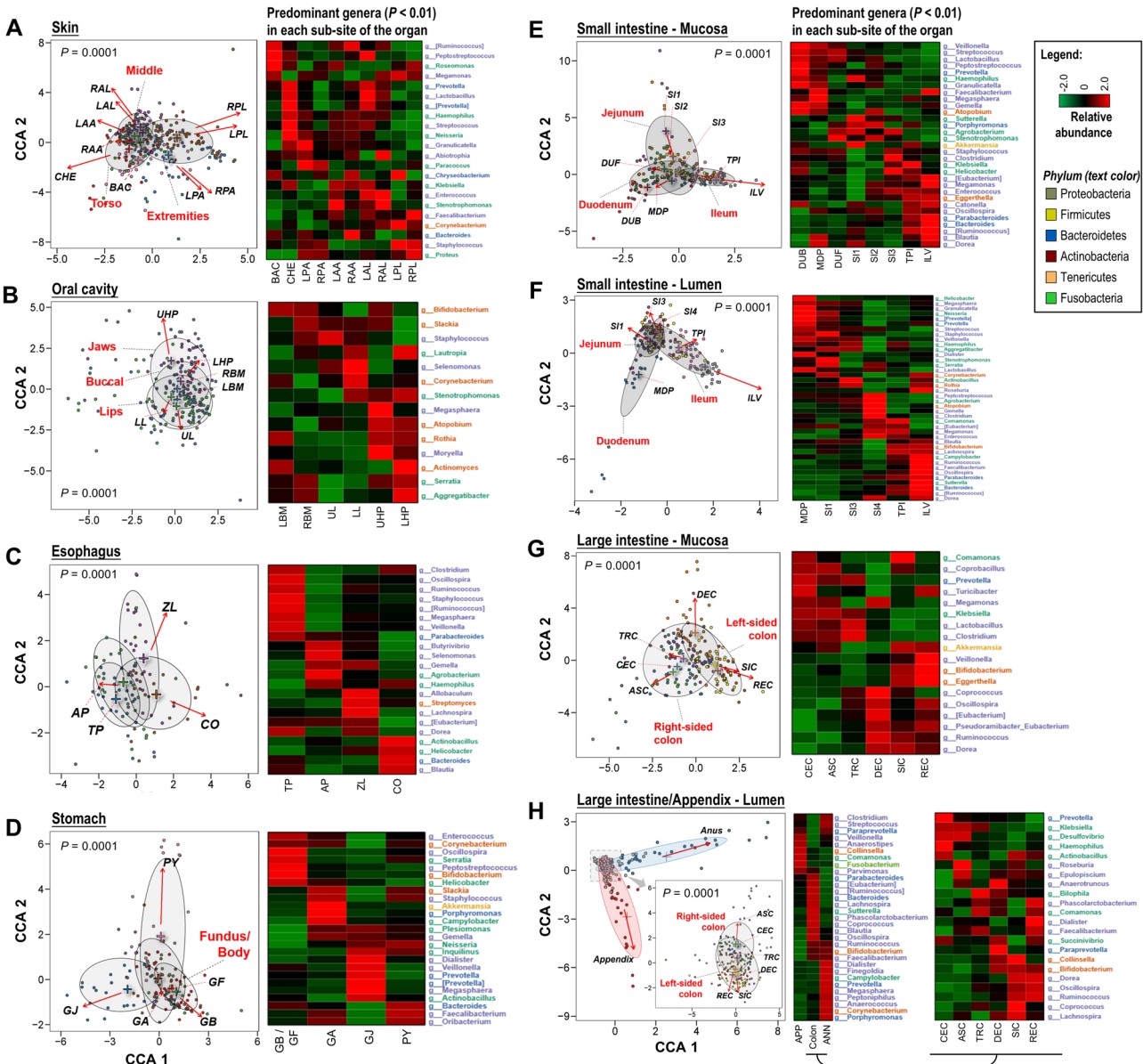

**Fig. 4 | Differentially enriched microbes in the intra-organ sites.** Intra-organ microbial communities were displayed in the *left* panel, measured using Constrained Correspondence Analysis (**A** skin; **B** oral cavity; **C** esophagus; **D** stomach; **E** mucosa of small intestine; **F** lumen of small intestine; **G** mucosa of large intestine; **H** lumen of large intestine and appendix). PERMANOVA analysis (adjusting for age, sex, BMI) was applied to test the significance of community dissimilarities. The arrow pointed to the direction of most rapid change towards the corresponding site. Differentially enriched microbes among intra-organ sites were displayed on the *right* panel. Selected microbes were colored based on their phyla. Source data are provided as a Source Data file.

## Microbial community differences between lumen and mucosa

The availability of paired lumen-mucosa samples allowed us to investigate the microbial difference between the two sample types. Using 16S v3v4 dataset, mucosal microbial communities were all significantly different from luminal microbial communities in stomach, small intestine and large intestine ($P < 0.0001$ for all, PERMANOVA) (Figs. 2B and 5A). To decipher the microbial relationships between lumen and mucosa, we used logistic regression and identified 33, 52, and 47 mucosal/luminal-associated microbes in stomach, small intestine and large intestine, respectively. In the stomach, 60% (9/15) of mucosal-enriched genera were members of *Firmicutes*; whilst major gastric juice-enriched microbes belonged to *Firmicutes* (47%, 7/15) and *Proteobacteria* (47%, 7/15; e.g., *Helicobacter*) (Fig. 5B). In the small intestine, *Firmicutes* occupied 50% (19/38) of mucosal-enriched microbes (e.g., *Coprococcus* and *Clostridium*), whereas 43% (6/14) of luminal-enriched microbes belonged to *Proteobacteria* (Fig. 5C). *Akkermansia* and *Bifidobacterium*,

two beneficial microbes in humans, were also enriched in the intestinal mucosa. In the large intestine, 81% (13/16) of mucosal-enriched microbes belonged to *Firmicutes* (Fig. 5D). Among luminal-enriched microbes, 42% (13/31) were members of *Firmicutes*, followed by *Bacteroidetes* (29%, 9/31). We then conducted similar analysis using 16S full-length dataset (Supplementary Fig. 3) in order to validate the above observations. We found that consistent lumen/mucosa-enriched bacteria were identified along the GI tract, including the stomach, small intestine, and large intestine (Supplementary Fig. 4). Moreover, in both small intestine and large intestine, we observed nine mucosal-enriched genera and seven luminal-enriched genera that are mucosal- and luminal-associated microbes (Fig. 5E), respectively.

## Functional capacities of microbiome differ among organs

Microbial functional attributes in surface organs were analyzed. Different pathways with significant enrichment were identified in each

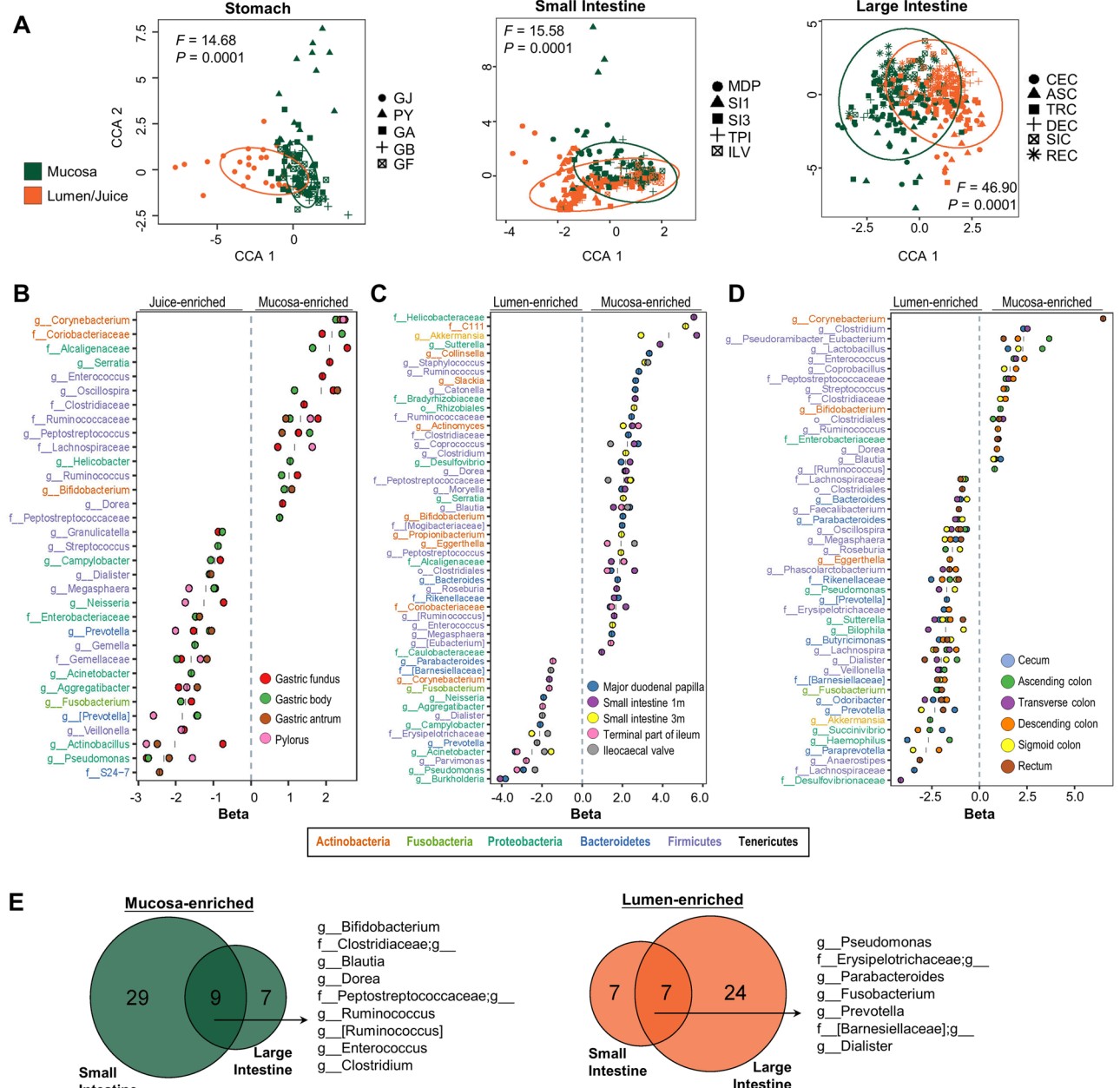

**Fig. 5 | Association of microbial niches with mucosa or lumen. A** Microbial dissimilarities between mucosal and luminal samples of the stomach, small or large intestines, measured using Constrained Correspondence Analysis. PERMANOVA analysis was applied to test the significance of mucosal samples compared to luminal samples. Each point represented an individual sample and was colored by sample types (mucosal or luminal sample) and shaped according to its originated site of the surface organ. Significant mucosa-enriched and lumen/gastric juice-enriched microbes in different sites of (**B**) stomach, (**C**) small or (**D**) large intestines, measured using logistic regression model. Beta values represented the magnitude of difference in relative abundance between paired luminal and mucosal samples, and the degree of consistency among subjects. Points were colored by sites. Selected microbes (FDR < 0.05) were colored based on their belonged phyla. **E** Shared mucosa-enriched (*left*) or lumen-enriched (*right*) microbes between the small intestine and large intestine. Source data are provided as a Source Data file.

organ (Supplementary Fig. 5A and Supplementary Data 1): aerobic respiration in the skin; nucleoside and nucleotide biosynthesis/degradation (e.g., adenosine and guanosine) in the oral cavity; fatty acid metabolism (e.g., gondoate biosynthesis) in the esophagus, stomach, and small intestine; and pentose phosphate pathway including glucose/sugars catabolism in the appendix and large intestine. Comparative analysis of metabolic pathways revealed several carbohydrates degradation pathways that are significantly enriched in the small intestine (e.g., sucrose degradation) and large intestine (*e.g.*, glycogen degradation of bacteria) (*P* < 0.05), respectively (Supplementary Fig. 5B). Amino acid synthesis (*e.g.*, L-isoleucine, L-aspartate,

L-histidine, and L-arginine) were significantly enriched in both the lower GI tract (appendix, small intestine and large intestine) and skin (*P* < 0.05) compared to other organs. Collectively, we revealed the differential microbial functional traits among surface organs.

## Intra-organ microbial interaction network reflects organ specificity

To uncover microbial interplay in each organ, we calculated pairwise microbial interactions in each organ using SECOM method (Supplementary Fig. 6). We observed that each organ has its own patterns of microbial interactions (Supplementary Fig. 7 and Supplementary

Data 2). Significantly different microbial interactions were observed among organs, with more co-exclusive relationships in oral cavity and large intestine, and more co-occurrent relationships in other organs (Supplementary Fig. 8A). We also used *SPARCC* method, which showed consistent findings (data not shown). Twenty-eight organ-specific microbial interactions was identified by both *SECOM* and *SparCC* (Supplementary Table 5), showing that microbial correlations were different among GI organs, for example, *Bacteroides* showed strong co-exclusive relationship with other microbes in the large intestine but strong co-occurrence with the same microbes in the upper GI organs (Supplementary Fig. 8B). These results implied that the microbiome in each organ habitat exhibits distinct microbe-environment relationships, suggestive of impact from host factors such as pH level and nutrient availability.

### Microbial inter-organ relations exist in GI organs

By sampling a large set of intra-individual sites, we attempt to characterize the microbial inter-organ relations (i.e., bacterial translocation) along the GI tract. We re-sequenced the samples using 16S full-length sequencing, which provides higher taxon resolution than 16S v3v4 region. We then measured the presence of bacterial ASVs (the exact sequence variants; relative abundance >0.1%) in the intra-individual organs using 16S v3v4 and full-length data, respectively. The ASVs were collapsed to species level and the species prevalence among individuals were calculated. Consistent results between the full-length sequencing and v3v4 region sequencing were found (Fig. 6A and Supplementary Fig. 9A). We discovered that oral pathogens (prevalence >50%; e.g., *Neisseria* spp. and *F. nucleatum*) were less prevalent in the GI tract (<50%), especially the lower GI tract (Fig. 6A). We then applied correlation analysis to indicate the co-enrichment or co-depletion of bacteria in multiple organs. We observed fewer bacteria with positive correlations ($P < 0.05$) between oral cavity and lower GI organs than that between oral cavity and upper GI organs (esophagus and stomach) (Fig. 6B and Supplementary Fig. 9B). These suggest that the oral-to-lower GI contribution is limited (5.5% ± 3.95% of oral bacteria) in healthy individuals. Moreover, Bacteria with positive correlations were more distinguishable within upper or lower GI organs (e.g., esophagus and stomach: ratio = 0.53; SI and LI: ratio = 0.51) than between upper and lower GI organs (e.g., esophagus and LI: ratio = 0.13) (Fig. 6B), supporting the evidence for the restricted bacteria translocation from the upper GI to lower GI organs in healthy individuals.

Some high prevalent bacteria in an organ were also prevalent (>50%) in other organs as shown in Fig. 6A. We therefore asked if these bacteria were simultaneously present (relative abundance > 0.1%) in the upper GI or lower GI organs from the same individual (core microbial species, defined as species that coexisted in different organs of the same individual). Indeed, there were ASVs co-existed in all upper GI or lower GI organs intra-individually (Fig. 6C), which was independently verified by 16S v3v4 data (Supplementary Fig. 10A). On the other hand, unique bacterial signatures were found in the upper GI or lower GI tract. For example, *S. salivarius* and *H. pylori* in the upper GI, and *Bacteroides* spp. (e.g., *B. vulgatus* and *B. caccae*) and *R. gnavus* in the lower GI. Shared signatures including *E. faecium*, *K. pneumoniae*, and *Enterobacteriaceae spp.* (*E. coli*, *E. flexneri*, and *E. sonnei*) were also found between the upper GI and lower GI tract (Fig. 6C). Moreover, correlation analysis confirmed their inter-organ relations in the lower GI and upper GI organs, respectively (Supplementary Fig. 10B). Our result thus suggests a microbiome core with significant inter-organ relations co-existed in different organs of the intra-individual GI tract.

### Discussion

In this study, 1608 surface/mucosal and luminal samples were collected from 53 distinct sites of seven surface organs per subject (skin, oral cavity, esophagus, stomach, small intestine, appendix, and large intestine). To our knowledge, we provided a much denser sampling of each individual along the digestive tract and skin than previous publications[7,11]. Using 16S full-length data to provide taxon information at species level, we identified the microbiome core species with significant inter-organ relations in the human GI tract. Taxa with high prevalence in ≥2 distinct organs such as *Streptococcus* were reported[4,12], of which the species of this genus was also identified in our samples (e.g., *S. salivarius*). Meanwhile, some of our core species were recognized as signature taxa in previous organ-specific microbiome studies including *R. gnavus* and *B. vulgatus* in the lower GI organs[13,14]. Hence, our results indicate that microbes with wide-acknowledged organ specificity could also be present in different body habitats, highlighting the importance of simultaneous microbial profiling in multiple organs to obtain a more comprehensive mapping of the human microbiome.

While the core species could co-exist in multiple organs, our results revealed that these species are specifically enriched in some organs (Fig. 3). Moreover, we analyzed differences in the microbial community among sites of an organ and revealed the intra-organ site-specific microbes. Across the skin sites, we identified the enrichment of *Corynebacterium* and *Staphylococcus*, of which these microbes are lipophilic and capable of utilizing sweat as nutrient source[15]. When comparing functional characteristics among surface organs, we reported the domination of aerobic respiration in the skin, which could be solely attributed to the reduced capability of gut anaerobes in aerobic respiration. The oral cavity is known to harbor a diverse microbiome as it comprises of hard non-shedding surfaces of the teeth and epithelial surfaces of the mucosal membranes[16]. We revealed that the oral mucosal microbiome could be clearly separated into distinct clusters including jaws/hard palates, buccal and lips with the respective enrichment of *Neisseria*, *Peptostreptococcus,* and *Staphylococcus* (Fig. 4), consistent with previous studies[4,13,17]. As the oral microbiome is exposed to various environmental factors including diet and living habits, its diversity is readily influenced by host behaviors[18]. The microbiome in the upper GI tract is less diverse since microbial growth is heavily limited by gastric acidity and peristalsis. Only a few microbes, particularly *Helicobacter* and *Lactobacillus* survive such harsh conditions[3]. Moreover, our functional analysis revealed significant enrichment of fatty acid metabolism in the upper GI tract, inferring the contribution of gut commensal microbes to dietary fat digestion by gastric lipase. Microbiome diversity reaches the top in the lower GI tract. *Streptococcus* and *Lactobacillus* were enriched in the duodenum, and these microbes are involved in primary bile acids deconjugation to balance lipid and carbohydrate metabolism[19]. In the jejunum, *Oscillospira* was found to be enriched which is a producer of butyrate, a crucial metabolite for maintaining gut homeostasis and energy metabolism[20]. In the ileum, *Bacteroides, Klebsiella,* and *Clostridium* were enriched, and these genera contribute to bile acids recycling and re-entry into enterohepatic circulation[19,21]. Owing to the easier accessibility, multiple studies have reported the microbiome heterogeneity between right-sided/proximal and left-sided/distal colon in healthy subjects and patients[22]. Consistently, we identified the enrichment of butyrate producers including *Klebsiella, Enterococcus,* and *Lactobacillus* in the right-sided colon, the major site of fermentation and microbes-mediated metabolism. *Parabacteroides, Bifidobacterium* and *Dorea* were enriched in the left-sided colon which is responsible for regulating gut motility[3]. Additionally, as the primary site for nutrient absorption, enrichment of multiple metabolic pathways including amino acid synthesis, carbohydrate metabolism and energy production was observed in the lower GI tract. Overall, our findings provide solid evidence of the association of microbial taxa with key physiological functions in a site-specific manner.

We collected paired mucosal and luminal samples from the GI tract. To date, most reports studied the mucosal microbiome by collecting endoscopic biopsies. Whereas some studies retrieved endoscopic

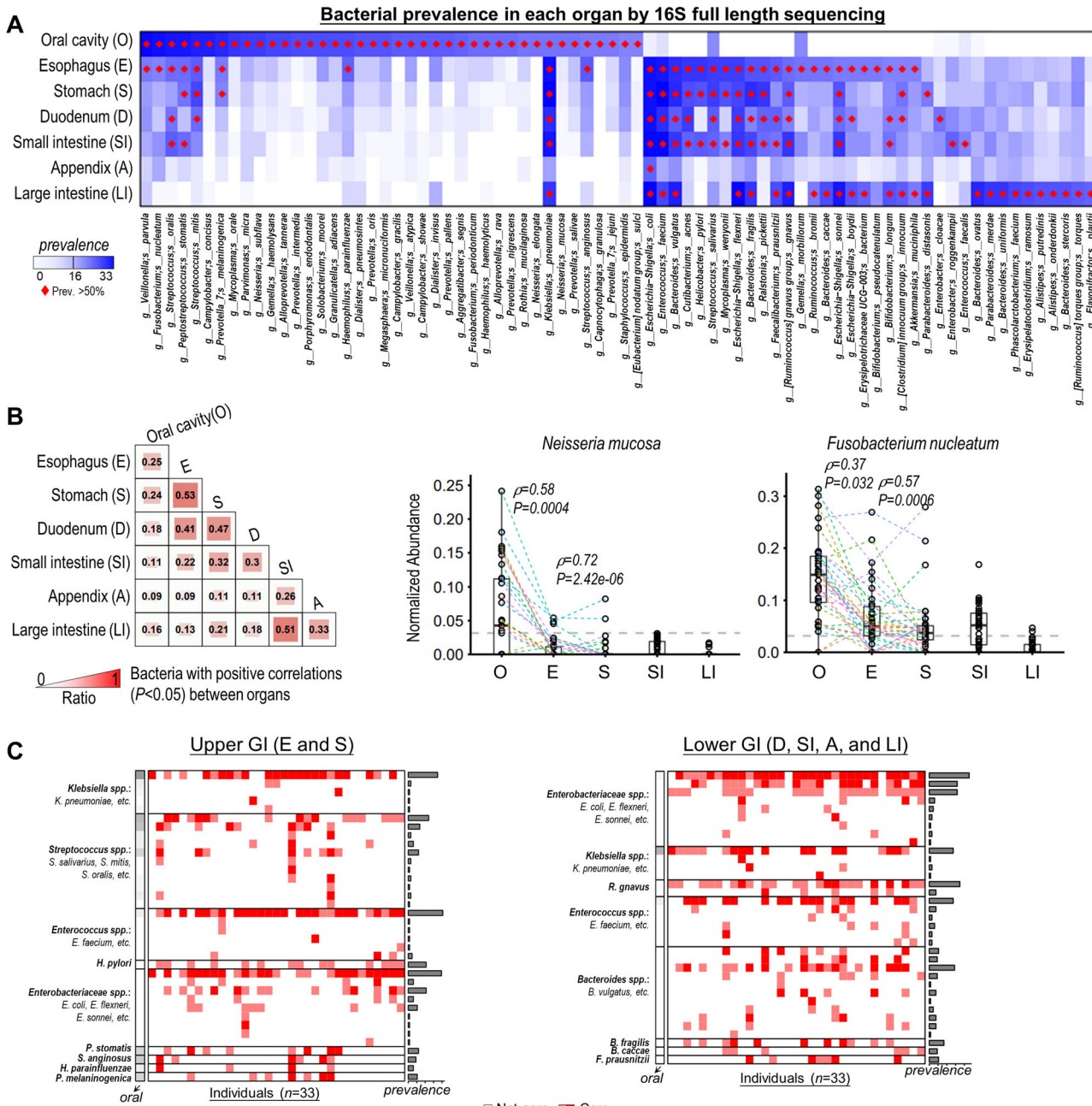

**Fig. 6 | Microbial inter-region relations along the GI tract. A** Bacterial prevalence in each organ by 16 S full-length sequencing. ASVs with relative abundance >0.1% (~10 sequencing reads) were considered as present on the organ. Red dot represents >50% prevalence. **B** The ratio of bacteria with positive correlations between each pair of organs among the prevalent bacteria (*n* = 76). The abundance of *N. mucosa* and *F. nucleatum* were plot in the right-side, with dash line links the same individual (*P* < 0.05, correlation analysis). Data are shown as Box and whisker plots to represent the median (center line), quartiles (box), range (whiskers), and outliers (points outside 1.5 times the interquartile range). Two-tailed Spearman correlation,

Partial Spearman correlation, and two-tailed Pearson correlation were used simultaneously. **C** ASVs simultaneously present in the intra-individual upper GI (*left*) or intra-individual lower GI tract (*right*). Areas labeled in red represent the presence of ASV on all the organs from the same individual (relative abundance >0.1% for all). Light red color: one type of ASV of a particular species shared among organs from the same individuals. Dark red color: >1 ASVs of a particular species shared among organs from the same individuals. Bacterial prevalence in the oral cavity was displayed on left side of the plot. Source data are provided as a Source Data file.

aspirates or gastric juice samples to examine luminal microbiome particularly in the small intestine, which has thinner wall and lower tolerance for multiple biopsies than the large intestine[23]. Here, we confirmed that the mucosal and luminal microbiome are distinct as evidenced by their significant differences in α and β-diversities. *Firmicutes* dominated the mucosal microbiome of small intestine (e.g., *Staphylococcus*, *Ruminococcus*) and large intestine (e.g., *Clostridium*, *Lactobacillus*), as well as the luminal microbiome of large intestine. Whereas *Proteobacteria* was

enriched in the small intestine lumen. Only 20% and 18% (Jaccard index) of luminal-enriched and mucosal-enriched microbes were shared between small intestine and large intestine, respectively. Additionally, mucosal-enriched microbes account for 73% and 34% of total microbes in small intestine and large intestine, respectively. These results collectively reflect microbial heterogeneity between the small intestine and large intestine, which may be attributed to the differences in pH level, bile salt concentration, and mucin composition[3].

Owing to our large collection of intra-individual samples, changes in microbiome diversity along intra-individual surface organs could be assessed. We discovered a gradual shift in microbial diversity along the GI tract (Fig. 2C). Microbial changes could be attributed to environmental factors (of each organ) and adaptation of microbes per se[3]. It is therefore of interest to assess the microbial flow along the GI tract. Compared to OTU-based analysis, ASV-based analysis offers advantages such as a finer resolution down to the level of single-nucleotide differences[24]. A single base difference in the 16S sequence will result in a unique ASV, thus providing a more detailed profiling of microbial diversity. Here, the ASVs with the exact sequence variant co-existed in different intra-individual organs. In addition, the inter-organ correlation of these ASVs revealed the species co-enrichment or co-depletion in multiple organs, indicative of the inter-organ bacterial translocation. Interestingly, we identified that the bacteria with significant inter-organ relations were clustered in the upper GI or lower GI organs; whilst only a few bacterial cross-contact between organs of the upper GI and lower GI were found, suggesting that the bacterial translocation were restricted between the upper GI and lower GI, thus explaining the limited contribution of oral microbiome to lower GI in healthy individuals. These results provide evidence that microbes in different organs could be carried by the luminal flow and accumulate in other organs, but could be constrained by the environmental factors in an organ-specific fashion.

In conclusion, we have generated a comprehensive biogeographical microbial mapping of seven human surface organs. This allowed elucidation of microbes that are present in multiple organs or in a particular organ. We also revealed the microbiome in different sites within each surface organ, and linked these site-specific taxa to key functional characteristics. In addition, we explored crosstalks of microbiome among different organs by analyzing microbial inter-region relations along the GI tract. Overall, our discoveries enhance our current understanding of the human microbiome by unraveling the details of various features of the microbial communities in surface organs.

## Methods

### Human subjects
All the human donors were declared dead by cardiovascular death. Causes of death can be found in Supplementary Table 1. Subjects were excluded if they had tumour, infectious disease, or metabolic disease. Written informed consent was obtained from each enrolled donor via next-of-kin to permit the collection and banking of samples (consent rate: 24.2% (33/136)). Sample collection was conducted under the instruction and supervision of Organ Procurement Organization of First Hospital of Xi'an Jiaotong University and the Red Cross Society of Shaanxi Province (Supplementary Methods). Samples from multiple surface organs were collected right after the harvest of liver and kidney for organ transplantation in the First Affiliated Hospital of Xi'an Jiaotong University. Characteristics of recruited 33 subjects were provided in Supplementary Table 1. The study was approved by the Clinical Application Ethics Committee of First Affiliated Hospital of Xi'an Jiaotong University (Approval No. XJTU1AF2019LSK-059) and conducted in accordance with the Declaration of Helsinki.

### Surface organ collection
The short duration (<1.5 h) of sample collection from the donor after death permits a thorough survey on microbial community of all human surface organs. Previous reports suggested that sampling within 2 h after death did not cause significant post-mortem changes in microbiome[25–27]. Surface (swab/mucosa) samples and luminal samples of 53 sites from seven surface organs were collected per individual in the 100-level laminar operating room. Each sample was collected using disposable surgical instruments to avoid any external bacterial contamination, instrument-related contamination, and cross-site

contamination. In addition, we parallelly introduced a set of environmental negative controls to measure the effect of laboratory-borne contamination (Supplementary Fig. 1A) during sample collection. These control samples were used to determine taxa that arise from contamination. All samples were frozen immediately using dry ice or liquid nitrogen and stored at −80 °C within 1.5 h for long-term storage.

### Detailed collection protocols from individual surface organs of human subjects
Samples from the oral cavity and skin surface were collected first. Intestinal mucosal and luminal/surface samples were collected in sequence: esophagus (4 sites), stomach (5 sites), small intestine (14 sites), appendix (1 subsite) and large intestine (13 sites).

**Skin.** Ten skin sites representing a range of physiological characteristics were selected, including core/proximal body sites: chest, back, palmar (left and right), plantar (left and right), anterior leg (left and right), and anterior antebrachium (left and right). Skin samples were collected with wet cotton swabs soaked in sterile saline by wiping repeatedly more than 30 times.

**Oral cavity.** Oral samples were obtained by rubbing the buccal mucosa (left and right), hard palate (upper and lower), and inside of lips (upper and power) with wet cotton swabs soaked in sterile saline by wiping repeatedly more than 30 times. Samples were collected without touching the teeth to avoid contamination by microbes present on the tooth surface.

**Esophagus.** Thoracic esophagus was pulled carefully down to the peritoneal cavity through esophageal hiatus. A 1–1.5 cm longitudinal incision was made in the anterior wall to expose the mucosa of esophagus. Mucosal samples were collected from the thoracic part, abdominal part, zigzag line, and cardiac orifice using disposable surgical scissors and forceps. The incision was sutured immediately after sample collection.

**Stomach.** Gastric juice was retrieved before the collection of mucosal samples. Mucosa specimens were collected from four sites (cardia, fundus, antrum, and pylorus) according to their anatomical locations.

**Small intestine, appendix, and large intestine.** The sample collection procedure was as follows: 1) Small intestine: 9 sites including duodenal bulb, major duodenal papilla, duodenojejunal flexure, small intestine 1 m, small intestine 2 m, small intestine 3 m, small intestine 4 m, terminal part of ileum, and ileocaecal valve; 2) Appendix; 3) Large Intestine: 7 sites including cecum, ascending colon, transverse colon, descending colon, sigmoid colon, rectum, and annus. For each subsite of small and large intestines, luminal samples were retrieved before the collection of mucosal samples. Luminal samples were quickly transferred to a sterile 50 ml centrifuge tube (BD, USA). Mucosal samples were rinsed gently with sterile saline to avoid being contaminated by intestinal contents.

### DNA extraction
Ultraclean kits and reagents were used to avoid exogenous DNA contamination. Mucosal samples (25–30 mg) were disrupted by bead-beating and digested in an enzymatic cocktail of mutanolysin and lysozyme (Sigma, St. Louis, MO) prior to DNA extraction with QIAamp DNA Mini Kit (Cat No.51306, Qiagen, Hilden, Germany). Swab sample was dissolved in a 2 ml RNase-free tube (Biosharp, China) with 500 μl sterile PBS and DNA was extracted using QIAamp DNA Mini Kit (No.51306, Qiagen). DNA from lumen samples was extracted using QIAamp Fast DNA Stool Mini Kit (No.51604, Qiagen). The negative control samples underwent identical processing procedures.

## 16S ribosomal RNA (rRNA) gene sequencing

The v3v4 regions of 16S rRNA genes were amplified using primers 341F [5′-CCTAYGGGRBGCASCAG-3′] and 806R [5′-GGACTACNNGGGTATC-TAAT-3′] together with the adapters and barcode sequences, allowing directional sequencing covering the hypervariable region (Novogene, Nanjing, China). Sequencing libraries were generated using TruSeq® DNA PCR-Free Sample Preparation Kit (Illumina, USA), and sequenced on an Illumina NovaSeq platform (dual-index) to generate 250 bp paired-end reads.

## PacBio 16S rRNA gene full-length HiFi sequencing

The full-length 16S rRNA genes were amplified by PCR using primers 27F [5′-AGRGTTTGATYNTGGCTCAG-3′] and 1492R [5′-TASGGHT-ACCTTGTTASGACTT-3′]. PCR products were purified using AxyPrep DNA Gel Extraction Kit (Axygen Biosciences, USA). Amplicon pools were prepared for library construction using the Pacific Biosciences SMRTbellTM Template Prep kit 1.0 (PacBio, USA) and sequenced on PacBio RS II (LC-Bio Technology Co., Ltd., Hangzhou, China).

## Sequence curation and annotation

For Illumina 16S v3v4 sequencing data, raw paired-end reads of 16S rRNA gene sequence were quality-filtered and analyzed using QIIME2 (version 2019.4.0; default parameters) software[28]. Deblur algorithm was applied to reduce sequencing errors and dereplicate sequences with default parameters. Before dereplicating sequences that encoded amplicon sequence variants (ASVs), paired reads were joined and trimmed to 380 base pairs. After filtering chimera sequences, dere-plicated sequences were classified taxonomically using Greengenes database at 99% identity cut-off. ASVs detected in negative controls were eliminated.

For PacBio 16S full-length sequencing data, circular consensus sequence (CCS) reads were generated from raw subreads by SMRT Link (v6.0). CCS reads from different samples were distinguished by lima (v1.7.1). Primers of CCS reads were trimmed by cutadapt (v1.9; default parameters) to obtain the clean reads. The clean reads with length between 1200 bp to 1650 bp were kept for further analysis. DADA2 algorithm was applied to dereplicate the reads and filter chimeric sequences. The dereplicated sequences (ASVs) were classified taxonomically using Greengenes database, SILVA database, and NCBI database by BLAST tool kits. ASVs detected in negative controls were eliminated.

Microbial community analysis, including α-diversity and β-diversity, were calculated using phyloseq R package. α-Diversity was evaluated by relative inverse Simpson index. β-iversity was measured by UniFrac distance, and principal coordinates analysis (PCoA) was used for ordination analysis. We applied two-tailed Wilcoxon signed-rank test for differential testing of α-diversity, and $P < 0.05$ was considered statistically significant. Community dissimilarities were tested by per-mutational multivariate analyses of variance (PERMANOVA) with 10,000 iterations. Differentially enriched microbes were analyzed using ANCOM-BC2 (v2.2.2; default parameters), a methodology for performing differential abundance (DA) analysis of microbiome count data[29]. Differences with fold change >2 and adjusted $P < 0.05$ were considered statistically significant. We applied constrained corre-spondence analysis to evaluate microbial dissimilarities of intra-organ sites and that of the lumen and mucosa.

## Controls for contamination during all stages of experiment

Several procedures were conducted to control for false-positive con-taminating taxa. We reduced the read counts to 10,000 library size to reduce the variation of library depths of each sample. Taxa with rela-tive abundance <0.1% in all samples were discarded, as their inclusion could introduce noise variation. We finally compared the taxa pre-valence of real biological samples to that of negative controls using decontam method[30] (threshold = 0.5), commonly used to discriminate true positives and contaminations.

## Luminal- and mucosal-related microbes by logistic regression

We applied a logistic binomial regression with overdispersion to unravel luminal- or mucosal-related microbes:

$$\log \frac{p}{1-p} = \beta_0 + \beta_2 R_{type} + \beta_k S_1 + \beta_{k+1} S_2 + \cdots$$

$$C \sim Binomial\,(N, p)$$

Where $p$ represents the probability of observing taxa $x$ after account-ing for the overdispersion using beta distribution $p \sim Beta\,(a, b)$. The hyperparameters $a$, $b$ were estimated automatically using the aod R package. $R_{type}$ and $S_i$ represent indicator variables for the sample type (lumen and mucosa/gastric juice) and human characteristics, respec-tively. $C$ represents the read count of taxa $x$ observed in this site, while $N$ indicates the sequence depth of all taxa observed in this sample.

## Functional analysis

Functional attributes of the microbiome associated with different surface organs were analyzed by PICRUSt2. Functional attributes were annotated by MetaCyc database, and the super-class of pathways was obtained using Pathway Tool (ver. 25.5). Differentially enriched path-ways among organs were analyzed using ALDEx2 method, as sug-gested by PICRUSt2 pipeline. Differences with fold change >2 and FDR < 0.05 were considered statistically significant.

## Correlation network analysis

SECOM method[31] was used to infer microbial interplay and was applied separately to each site. We also applied the SparCC method[32] to further validate the observations. Microbial correlations were selected if they have FDR < 0.05 and shared the same correlation type (positive/negative correlation) in all sites of an organ, and the average correla-tion was calculated. We then defined the organ-specific correlation, which met any of the following two conditions: 1) the difference in correlations between organs >0.6; 2) the correlation with strength >0.6 was present in this organ only. The selected correlations were visualized using Cytoscape (version 3.7.1).

## Statistical analysis

Statistical significance tests, including Wilcoxon signed-rank test, ANOVA permutation test, ANCOM-BC2, SECOM, and SparCC correla-tion test, were performed using open-source R software. All statistical tests were two-sided, and $P < 0.05$ were considered statistically sig-nificant. For multiple comparisons, $P$ values were adjusted using Benjamini-Hochberg False Discovery Rate (FDR) correction.

## Reporting summary

Further information on research design is available in the Nature Portfolio Reporting Summary linked to this article.

# Data availability

All the raw sequencing data generated in this study have been deposited in NCBI Sequence Read Archive (SRA) under BioProject PRJNA1049979. ASV sequences were classified taxonomically using Greengenes database at 99% identity cut-off. Remaining data are available within the Article or Supplementary Information. Source data are provided with this paper.

# Code availability

Source code and scripts performed for the study have now been uploaded to: https://github.com/WilsonYangLiu/DCD-bacteria.git.

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

## Acknowledgements

This study was supported by National Natural Science Foundation of China (81870380/82173394/82100554/82303941), Key Research and Development Program of Shaanxi Province (2020ZDLSF01-03), National Key R&D Program of China (No. 2020YFA0509200/2020YFA0509203), China Postdoctoral Science Foundation (2019M663748), Natural Science Basic Research Program of Shaanxi (S2023-JC-QN-2665), Integrative "Basic-Clinical" Innovation Program (YXJLRH2022043), International science and technology cooperation program of Shaanxi Province (2020KWZ-020), RGC Research Impact Fund (R4017-18F), RGC Theme-based Res Scheme Hong Kong (T21-705/20-N).

## Author contributions

J.J.S. collected the samples and managed the study. W.L. performed all the computational analysis and drafted the manuscript. X.M.D. collected the samples. G.G. collected the samples, performed experiments, and revised the manuscript. J.H. provided bioinformatic support and revised the manuscript. H.C.H.L. and C.C.W. revised the manuscript. F.Y.S., C.G.D., W.J.X., W.S., G.X.L., Z.Z., C.H.H. and Y.C. collected the samples and performed DNA isolation. J.Y. designed, supervised the study and revised the manuscript.

## Competing interests

The authors declare no competing interests.
