## [Peer Review File · Nature Communications]

Reviewers' comments:

Reviewer #1. Biostats, microbiome work. (Remarks to the Author):

Reviewer's comments on "Defining the biogeographical map and potential bacterial translocation of microbiome in human surface organs"

Using biospecimens obtained from 33 human donors who passed away due to brain or cardiovascular disease, the authors developed the microbiome atlas in human body. All samples were retrieved within 90 minutes after death. This is an interesting study and I commend the authors for conducting such a study. I have several comments listed below.

1. Although the authors have not included donors whose death was due to cardiovascular disease, did they make sure the subjects did not suffer from any of the components of the metabolic syndrome, HIV, or GI related diseases? Some donors were as young as 34. What was the cause of death? Perhaps Table S1 should include cause of death and any disease information they may have. Since microbiota are involved in so many diseases, it is hard to interpret the results derived in this study.

(a) Include in supplementary table a column describing comorbidities/cause of death. There may be many different comorbidities/causes of death among the 33 subjects. In such a case, try to group them into fewer distinct subsets, e.g., accidental death/homicide, use of drugs (e.g., overdose of a prescription drugs/opioids) etc. and then perform either a formal analysis of covariance or not perform any formal analysis of covariance but in the discussion section, qualitatively discuss/interpret the results taking into account the comorbidities/cause of death. Perhaps draw on some existing literature to interpret the results. This could potentially generate interesting hypotheses for future research.

2. The alpha diversity results (Figure 2) were surprising to me. If I am reading correctly, there is no significant difference between stomach and the large intestine. I expected the gut to have the largest alpha diversity among all sites. Was the alpha diversity calculated at the phylum level or species level?

3. Relative abundances by various sites are presented in Figure 3. I am not sure how to interpret those figures. For example, the density plots of each taxon in Figure 3 A1 need to be interpreted in the context of other taxa. Perhaps the authors should use stack bar graphs as often by researchers presenting microbiome distributions. How were the p-values presented in Figure A2 derived? These are relative abundances that are being compared across sites. There are several methods available in the literature to do this such as LOCOM, LiNDA, ANCOM-BC2 etc. Similarly, how was the heatmap in Figure B constructed. Note that standard concepts of correlation coefficient for microbiome data are not valid. Did the authors use SparCC when constructing the heatmaps?

4. Comments similar to #3 above hold for several other figures in the paper.

5. The authors used SparCC for computing correlations between taxa. As it was recently demonstrated, SparCC is subject to poor sensitivity and specificity. In fact, the method performs poorly when computing correlations across multiple sites. The authors should instead consider SECOM (Lin et al., 2022).

Please discuss Cathy Lozupone's (2012) paper that discusses variations in the microbiome colonies at different body sites. I think that article is potentially relevant:

- Lozupone CA, Stombaugh J, Gonzalez A, Ackermann G, Wendel D, Vázquez-Baeza Y, Jansson JK, Gordon JL, Knight R. Meta-analyses of studies of the human microbiota. *Genome Res.* 2013 Oct;23(10):1704-14. doi: 10.1101/gr.151803.112. Epub 2013 Jul 16. PMID: 23861384; PMCID: PMC3787266.

Reviewer #2. Bioethics and organ collection (Remarks to the Author):

I cannot comment on the scientific merits of the research as this is beyond my expertise, however I would be interested to know why the analysis does not consider the potential influence of end-of-life care experiences and donation pathways (after circulatory or neurological determination of death) on the microbiome results. Further, while there is reference to consideration of donor sex, only 2 of the 33 participants were female and it's not clear what implications this might have for the results. The supplementary materials also refer to exclusion of non-Chinese donors in order to ensure homogeneity in the sample population, but there is no mention of ethnicity which is distinct from citizenship. I appreciate that Shaanxi province is predominantly Han Chinese, but China notably has more than 50 official ethnic minorities. Please clarify if all participants were resident specifically of Shaanxi province, and if any ethnic minorities were included.

Some more detail regarding the recruitment, donation and ethical aspects of the protocol would be helpful, especially in the light of persisting international concerns regarding the ethical procurement of organs and tissues after death in China. Some of these details might be provided in a supplementary document whereas others may be considered sufficiently important to include in the main manuscript.

The authors may find it helpful to consider the example of Sharma et al. (2023) which describes a process of donor recruitment and consent for use of samples in research:
<https://www.ncbi.nlm.nih.gov/pmc/articles/PMC9750700/>

Specifically, please outline when, how and by whom consent for collection of samples was obtained from the donors' next-of-kin. Please also make clear the earlier process of consent for donation of organs for transplantation purposes, given the considerable variation in policy and practices within China with regards to donation decision-making protocols. For example, was it the donation coordinators who sought consent for further donations for use in this study or was this a separate process? What was the local consent model in place, e.g., is it opt-out? Explicit consent? Were the actual donors individuals who

had previously volunteered to join the organ donor registry? Is a reward of financial value offered to families who provide consent for donation? Was any further reward or compensation offered to those who agreed to consent to sample collection for use in research? What was the consent rate for participation amongst families approached regarding this study?

Please provide some clarification regarding the disproportionate number of males recruited for this study. Was consent lower among families of potential female donors? More demographic details regarding the circumstances of the donors' deaths would be helpful and may be pertinent in analysis of the results. For example, what was the duration of length of stay in intensive care prior to death? What was the duration between a decision to cease life sustaining interventions and commencement of a DCDD pathway? Between determination of death by neurological criteria and commencement of organ procurement? What was the primary cause of death in these cases? When you note that individuals with "metabolic disease" were excluded, does this mean all those with comorbidities such as diabetes were excluded?

Finally, please confirm explicitly that no organs or other specimens used in this study were obtained from executed prisoners.

Reviewer #3. Human microbiome (Remarks to the Author):

The study focused on examining the microbiome in various human organs, specifically investigating the microbial communities within and between these organs. The authors collected samples from different regions of seven surface organs (oral cavity, esophagus, stomach, small intestine, appendix, large intestine, and skin) from 33 subjects, totaling 1608 samples. They used 16S rRNA sequencing techniques to analyze the microbial diversity and species and used PICRUST to infer the microbial functions. The findings of the study revealed significant variations in microbial diversity among different surface organs. They observed that certain core microbial species were consistent across different organs within the same person. The authors also delved into the microbial changes within specific regions of individual organs and identified signature microbial species, their functions, and interactions unique to each region. The study highlighted differences in the microbiomes between paired mucosa-lumen samples of the stomach, small intestine, and large intestine. Additionally, they constructed a comprehensive overview of the relationships between microbes across the digestive tract, establishing an inter-organ microbial landscape.

This study holds merit in its substantial sample size and the inclusion of samples from diverse body sites, which enhances its credibility. Nevertheless, a notable limitation lies in its heavy reliance on 16S data,

rendering it predominantly compositional and descriptive in nature. Furthermore, a significant portion of the study's conclusions lack novelty, as they align with existing knowledge.

Reviewer #4. Low density/ biomass microbiomes (Remarks to the Author):

In this study, the authors focused on profiling microbiomes found in human surface organs, including the oral cavity, gastrointestinal tract (GI), and skin. They collected 1608 samples from 53 regions of the surface organs. All samples were profiled with 16S v3v4 region sequencing. Among those samples, 1030 samples from GI organs were profiled with PacBio 16S full-length HiFi sequencing in order to be analyzed at high resolution (species rank). Based on the taxonomic profiling of the samples, the authors showed 1) the difference in microbial diversity, composition, and interaction in both inter-organ and intra-organ perspectives, 2) the difference in functional pathways among different organs, and 3) microbial community differences between lumen and mucosa samples in the same organs.

The experimental design and the analysis are sound and well-explained. The authors performed negative control experiments and multiple testing corrections to minimize the potential bias in their microbiome study. The findings offer a detailed view of human host-associated bacteria.

Major comments

1. In this study, the term "core microbial species" is mentioned in the abstract as species that coexisted in different organs of the same individual. Please also formally define this concept in the main text since it is a key concept of this study.

2. Given that short-read 16S sequencing of the V3-V4 region commonly does not provide sufficient taxonomic information to resolve taxonomy at the species level, the core microbial species are determined by PacBio full-length 16S sequencing, which only includes the GI organs. Is this correct?

3. In Figure 6, the authors state, "Areas labeled in red represent the presence of ASV on all the organs from the same individual." However, the core species were labeled in light red and dark red. Clarify the difference between light vs dark colors.

4. The authors need to formally describe the concept of upper GI and lower GI in the manuscript. The terms upper GI and lower GI were used across multiple sections, with seemingly inconsistent meanings.

For example, in Figure 2B, the definition of upper GI includes duodenum(D). However, in Figure 6, duodenum(D) was included in the lower GI instead. Does including duodenum in the upper GI changes your finding?

5. In Figure 2C, the author provided a PCoA plot of the beta diversity of microbiomes from different organs and showed distinguishable clusters between them. However, the authors state, "In the small intestine, we observed drastic intra-organ variation in microbiome spanning between the stomach and large intestine." Since the samples are collected from different regions of the organs, and intra-organ analysis is done in Figure 4, it would be very helpful to know if colored by different regions, whether clear clusters or beta diversity shifts can be observed based on the locations of sampling across organs. More specifically, the authors should show whether the intra-organ variation in the small intestine cluster that spans between the stomach and large intestine clusters you observed in Figure 2C is associated with the sampling location.

6. Figure 5A shows two different analyses. In the stomach, the comparison is between all lumen and mucosa samples without specification of the sampling regions. However, in the small intestine and large intestine plots, the comparison is between selected regions where you have both lumen and mucosa samples, and regions without both lumen and mucosa samples are excluded. This analysis should be made consistent to compare apples to apples.

7. Overall, the figure legends would benefit from additional details. Please indicate the meanings of colormaps for all heatmaps, including Figure 3B, Figure 4, Figure 6A (I'm guessing the red dot is 0.75 prevalence), and 6C (as mentioned before).

8. It was unclear which analyses were based on 16S full-length sequencing and which were based on 16S v3v4 sequencing. Is 16S full-length data just being used on inter-organ core species analysis (Figure 6)? For example, if that's true, since 16S full-length sequencing provides better classification accuracy and resolution, why not use those data to compare lumen and mucosa samples (Figure 5)? Does the conclusion still hold with 16S full-length sequencing data? Please clarify.

9. The data availability statement indicates all data is available via a genome sequence archive bioproject ID, but when accessing the link it returns "The data you are retrieving is not released in BioProject: PRJCA017513" and "Wrong share code OR this preview is Cancelled." so it seems the data are not yet made available?

10. Regarding all of the source code, scripts, and outputs from the analyses performed for the study, I was unable to locate them. Similarly, no parameters were indicated for the software used (qiime, cutadapt, decontam, etc).

11. The authors include ASV based analysis throughout, yet fail to include a discussion (or comparison to) OTU based analyses and approaches (for reference, please see: <https://www.ncbi.nlm.nih.gov/pmc/articles/PMC8870492/>). The scientific findings would be strengthened by a discussion of, or comparison to, OTU based approaches.

Minor comments

1. Figure 1A is unclear since not all samples were sequenced based on 16S full-length sequencing and requires clarification.
2. Please add the description for the top subfigure of Figure 2C in the figure legend.
3. Please be more specific on Figure A1; what is the x-axis label? I'm guessing individual samples. How were those samples clustered?
4. For Figure S2, fitted curves of relative abundance are mentioned in the figure legend but not visible in the figure.
5. Requires rewrite due to unclear meaning: "Moreover, we found that there are more bacteria with positive correlations between organs from the upper GI or lower GI than that of upper-to-lower GI organs (Figure 6B)" to clarify that the positive correlations are more distinguishable within upper/lower GI groups than across groups if this is indeed the case.
6. Figure S3A is confusing as the coloring is based on the superclasses of pathways. However, I'm unsure if the p-values are calculated based on the superclasses or finer categories. The superclass profiles do not look very different between organs; please clarify this.

Reviewer #5 (Remarks to the Author):

Reviewer #6. Low biomass microbiome study, contamination (Remarks to the Author):

Thank you for sending me the manuscript to review, “Defining the biogeographical map and potential bacterial translocation of microbiome in human surface organs”. She et al describe a multi-organ microbiome survey focussing on 53 sites in 33 individuals, and examine the signature species and shared taxa between them as well as the inferred functional differences.

It is to be applauded that the study undertakes much denser sampling of each individual than is generally found in the literature. However the authors seem to suggest (line 60, 262) that within-host translocation or microbial community changes along the gut have not been studied before, but this is not the case, for example [Vonaesch et al 2018, pubmed 30126990] specifically addresses pathological translocation between oral/stomach/duodenum/colon.

There is an important aspect of the methodology that is not mentioned in the main text: I was surprised to reach page 18 before discovering that the subjects were all deceased organ donors.

Although the authors state that as the subjects died very shortly before sampling there would not be any post mortem deterioration in the microbiome, there is no elaboration on the other associated factors that would affect the microbiome in the digestive tract and elsewhere (such as length of hospital stay, time spent in ICU, use of enteral feeding, surgical or antibiotic treatments that occurred before sampling). For example it is known that critically ill patients in hospital commonly experience reduced gut motility, which can lead to increased bacterial translocation, a specific focus of this manuscript.

It would be helpful for the authors to address this directly in the text, and state whether the results can confidently be generalised to the microbiome of non-hospitalised living people, and if not then where the caveats lie.

A note about the decontamination methodology: 5% of ASVs being removed by Decontam seems quite a large proportion, higher than I would expect from exogenous contamination sources in high biomass samples. However there is also a significant risk of inter-sample contamination (cross contamination during sample handling or PCR, as well as barcode bleed) which should not be wholesale removed.

Barcode bleed is greatly reduced by using dual-barcoding, but it is not clear from the methods whether single- or dual-indexing was performed.

The readers' confidence in the Decontam removal would be improved if the authors briefly describe what was removed - for example if they're derived from exogenous sources then the removed ASVs should include environmental taxa and would be expected to show a batch effect and make up a higher proportion in the lower biomass samples.

A final minor point, but there are a number of non-standard word choices. The main example is in the title, "surface organs", which makes sense conceptually once it is explained but does not mean what it first appears to (an external organ). Have the authors coined this phrase? If so please state that is the case or otherwise cite the original useage. It may reduce confusion in the title to add ".

Over all the study demonstrates a dense sampling strategy that allows novel insights to be drawn regarding the compartmentalisation (or non-compartmentalisation) of these organs' microbes. If the authors are able to address the points above I should be happy to see it published.

Many thanks,

Response to the comments of editors and the referees in relation to the manuscript NCOMMS-23-23427:

COMMENTS are written in *italics* and RESPONSES in normal text.

Response to Reviewer #1:

Reviewer's comments on "Defining the biogeographical map and potential bacterial translocation of microbiome in human surface organs" Using biospecimens obtained from 33 human donors who passed away due to brain or cardiovascular disease, the authors developed the microbiome atlas in human body. All samples were retrieved within 90 minutes after death. This is an interesting study and I commend the authors for conducting such a study. I have several comments listed below.

1. Although the authors have not included donors whose death was due to cardiovascular disease, did they make sure the subjects did not suffer from any of the components of the metabolic syndrome, HIV, or GI related diseases? Some donors were as young as 34. What was the cause of death? Perhaps Table S1 should include cause of death and any disease information they may have. Since microbiota are involved in so many diseases, it is hard to interpret the results derived in this study.

(a) Include in supplementary table a column describing comorbidities/cause of death. There may be many different comorbidities/causes of death among the 33 subjects. In such a case, try to group them into fewer distinct subsets, e.g., accidental death/homicide, use of drugs (e.g., overdose of a prescription drugs/opioids) etc. and then perform either a formal analysis of covariance or not perform any formal analysis of covariance but in the discussion section, qualitatively discuss/interpret the results taking into account the comorbidities/cause of death. Perhaps draw on some existing literature to interpret the results. This could potentially generate interesting hypotheses for future research.

RESPONSE: We have carefully checked the medical history of the donors and confirmed that all donors did not suffer from any of the components of the metabolic syndrome, HIV, or GI related diseases (**Table S1**). Causes of death are shown in **Table S1**, which are mainly caused by cerebral hemorrhage due to vehicle accident, high-altitude falling, etc. We have now performed permutational multivariate analysis of variance (PERMANOVA) to determine the effect of subject's characteristics (e.g., cause of death, length of hospitalization) on the microbiome communities. We found the length of hospitalization and antibiotic treatments had significant effects on microbiome in the oral cavity, small intestine, and large intestine, but not the cause of death (**Table S3**). We have now added this information to **Results** section of the revised manuscript, as follows:

Results (p.6, line 96-99):

We collected 1,608 samples from 7 surface organs of oral cavity (6 sites), esophagus (4 sites), stomach (5 sites), small intestine (14 sites), appendix (1 site), large intestine (13 sites) and skin (10 sites), in total comprising of 53 sites in 33

subjects (**Figure 1A**) who were dead due to vehicle accident, high-altitude falling, etc. (**Table S1**).

Results (p.6, line 111-116):

We next applied permutational multivariate analysis of variance (PERMANOVA) to study the effect of subject's characteristics (e.g., cause of death, length of hospitalization) on the microbiome communities. We found the length of hospitalization and antibiotic treatments had significant effects on microbiome in the oral cavity, small intestine, and large intestine, but not the cause of death (**Table S3**).

2. *The alpha diversity results (Figure 2) were surprising to me. If I am reading correctly, there is no significant difference between stomach and the large intestine. I expected the gut to have the largest alpha diversity among all sites. Was the alpha diversity calculated at the phylum level or species level?*

RESPONSE: As shown in **Figure 2A** and **Figure S1C**, alpha diversity was significantly different between the stomach and large intestine at phylum ($P < 0.001$), genus ($P < 0.001$), and species levels ($P < 0.001$).

3. *Relative abundances by various sites are presented in Figure 3. I am not sure how to interpret those figures. For example, the density plots of each taxon in Figure 3 A1 need to be interpreted in the context of other taxa. Perhaps the authors should use stack bar graphs as often by researchers presenting microbiome distributions. How were the p-values presented in Figure 3 A2 derived? These are relative abundances that are being compared across sites. There are several methods available in the literature to do this such as LOCOM, LiNDA, ANCOM-BC2 etc. Similarly, how was the heatmap in Figure B constructed. Note that standard concepts of correlation coefficient for microbiome data are not valid. Did the authors use SparCC when constructing the heatmaps?*

RESPONSE: We have now used stack bar graphs for presenting the data in **Figure 3A1** and **Figure S2A**. In **Figure 3A2**, we calculated the P values using Friedman's test, a non-parametric test that compares three or more paired groups. In **Figure 3B**, we compared the abundance of each genus using ALDEx2 method, a compositional-oriented method that infers the difference among groups and is commonly used in microbiome research [1, 2]. ALDEx2 is used in **Figure 3B**, this heatmap plots the differential abundance at genus level, but not the correlation among them.

4. *Comments similar to #3 above hold for several other figures in the paper.*

RESPONSE: We have now revised the other Figures according to your helpful suggestions to make them clearer.

Figure 1A:

We have indicated the samples for 16S v3v4 and/or 16S full-length sequencing in details.

Figure 2B:

We have highlighted duodenum as part of the small intestine according to the comment of *Reviewer #4*.

Figure 5A:

We have annotated the sites for gastric mucosa samples according to the comment of *Reviewer #4*.

Figure 3 legend (p.33, line 672-674):

(B) Genus with significantly different abundance among seven organs by ALDEx2 method. **The colormap represents the average bacterial abundance.**

Figure 4 legend (p.34, line 683-684):

Selected microbes (FDR<0.05) were coloured based on their phyla.

Figure 6 legend (p.35, line 700-703; p.35, line 710-712):

(A) Bacterial prevalence in each organ by 16S full-length sequencing. ASVs with relative abundance >0.1% (~10 sequencing reads) were considered as present on the organ. **Red dot represents >50% prevalence.**

(C) **Light red colour: one type of ASV of a particular species shared among organs from the same individuals. Dark red colour: >1 ASVs of a particular species shared among organs from the same individuals.**

5. *The authors used SparCC for computing correlations between taxa. As it was recently demonstrated, SparCC is subject to poor sensitivity and specificity. In fact, the method performs poorly when computing correlations across multiple sites. The authors should instead consider SECOM (Lin et al., 2022).*

RESPONSE: We applied *SparCC* to compute the correlations in each site ($n=53$ sites) and compared the results across multiple sites (**Figure S6**), rather than pool all the sites to analyses together. We thus used *SparCC*. We have now performed additional analysis using *SECOM* (**Table S7**) and compared the results with *SparCC* (**Table S8**). A total of 28 organ-specific microbial interactions were commonly identified by *SECOM* and *SparCC*, and we observed high consistency between the two methods (**Table S8**). We have now added this information to revised **Results** section, as follow:

Results (p.12, line 230-234):

Twenty-eight organ-specific microbial interactions was identified by both *SECOM* and *SparCC* (Table S8**), showing that microbial correlations were different among GI organs, for example, *Bacteroides* showed strong co-exclusive relationship with other microbes in the large intestine but strong co-occurrence with the same microbes in the upper GI organs (**Figure S8C**).**

Please discuss Cathy Lozupone's (2012) paper that discusses variations in the microbiome colonies at different body sites. I think that article is potentially relevant:

• *Lozupone CA, Stombaugh J, Gonzalez A, Ackermann G, Wendel D, Vázquez-Baeza Y, Jansson JK, Gordon JI, Knight R. Meta-analyses of studies of the human microbiota. Genome Res. 2013 Oct;23(10):1704-14. doi: 10.1101/gr.151803.112. Epub 2013 Jul 16. PMID: 23861384; PMCID: PMC3787266.*

RESPONSE: We have now discussed Cathy Lozupone's (2012) paper, as follows:

Discussion (p.14, line 279-280):

To our knowledge, we provided a much denser sampling of each individual along the digestive tract and skin than previous publications^{7, 11}.

Response to Reviewer #2:

1. I cannot comment on the scientific merits of the research as this is beyond my expertise, however I would be interested to know why the analysis does not consider the potential influence of end-of-life care experiences and donation pathways (after circulatory or neurological determination of death) on the microbiome results.

RESPONSE: We have now provided detailed information of end-of-life care experience (cause of death, antibiotic use, etc) and donation pathways (cardiovascular death) for each subject (**Table S1**) and analyzed their potential effects on the microbiota (**Table S3**) using permutational multivariate analysis of variance (PERMANOVA). We found the length of hospitalization and antibiotic treatments had significant effects on microbiome in the oral cavity, small intestine, and large intestine, but not the cause of death (**Table S3**). We have now added this information to **Results** section of the revised manuscript, as follows:

Results (p.6, line 96-99):

We collected 1,608 samples from 7 surface organs of oral cavity (6 sites), esophagus (4 sites), stomach (5 sites), small intestine (14 sites), appendix (1 site), large intestine (13 sites) and skin (10 sites), in total comprising of 53 sites in 33 subjects (**Figure 1A**) who were dead due to vehicle accident, high-altitude falling, etc. (**Table S1**).

Results (p.6, line 111-116):

We next applied permutational multivariate analysis of variance (PERMANOVA) to study the effects of subject's characteristics (e.g., cause of death, length of hospitalization) on the microbiome communities. We found the length of hospitalization and antibiotic treatments had significant effects on microbiome in the oral cavity, small intestine, and large intestine, but not the cause of death (**Table S3**).

2. Further, while there is reference to consideration of donor sex, only 2 of the 33 participants were female and it's not clear what implications this might have for the results. The supplementary materials also refer to exclusion of non-Chinese donors in order to ensure homogeneity in the sample population, but there is no mention of ethnicity which is distinct from citizenship. I appreciate that Shaanxi province is predominantly Han Chinese, but China notably has more than 50 official ethnic minorities. Please clarify if all participants were resident specifically of Shaanxi province, and if any ethnic minorities were included.

RESPONSE: As we only have two female donors (due to the lower incidence of unnatural deaths in females), we are not able to evaluate the effect of donor sex on the composition of microbiome. We confirmed that all donors were Han Chinese and local residents of Shaanxi province (**Table S1**).

3. Some more detail regarding the recruitment, donation and ethical aspects of the protocol would be helpful, especially in the light of persisting international concerns regarding the ethical procurement of organs and tissues after death in China. Some of these details might be provided in a supplementary document whereas others may be considered sufficiently important to include in the main manuscript.

RESPONSE: We have now added detailed information regarding recruitment, donation, and ethical aspects in **Methods** section and **Table S1**. Informed consent has been obtained from next of kin for all donors (**Data S1. Informed consent example**). We have strictly followed the Guidelines for organ donation after cardiac death in China (**Data S2. DCD Guidelines** and **Data S3. Approval system for living organ transplantation**). These are added to **Methods** as follows:

Materials and Methods (p.19, line 376-385):

Human subjects

All the human donors were declared dead by cardiovascular death. Major causes of death include vehicle accident, high-altitude falling, etc. (**Table S1**). Subjects were excluded if they had tumour, infectious disease, or metabolic disease. Written informed consent was obtained from each enrolled donor via next-of-kin to permit the collection and banking of samples (consent rate: 24.2% (33/136)) (**Data S1**). No organs or other specimens used in this study were obtained from executed prisoners. Sample collection was conducted under the instruction and supervision of Organ Procurement Organization of First Hospital of Xi'an Jiaotong University and the Red Cross Society of Shaanxi Province (**Data S2-3**).

4. *The authors may find it helpful to consider the example of Sharma et al. (2023) which describes a process of donor recruitment and consent for use of samples in research: <https://www.ncbi.nlm.nih.gov/pmc/articles/PMC9750700/>*

RESPONSE: We have followed this publication to provide a more detailed description of donor recruitment (as described above).

5. *Specifically, please outline when, how and by whom consent for collection of samples was obtained from the donors' next-of-kin. Please also make clear the earlier process of consent for donation of organs for transplantation purposes, given the considerable variation in policy and practices within China with regards to donation decision-making protocols. For example, was it the donation coordinators who sought consent for further donations for use in this study or was this a separate process? What was the local consent model in place, e.g., is it opt-out? Explicit consent? Were the actual donors individuals who had previously volunteered to join the organ donor registry? Is a reward of financial value offered to families who provide consent for donation? Was any further reward or compensation offered to those who agreed to consent to sample collection for use in research? What was the consent rate for participation amongst families approached regarding this study?*

RESPONSE: We have followed a standardized protocol for getting informed consent for sample collection from donor's next-of-kin (**Data S1. Informed consent example**). First, the DCD (Donation after Cardiac Death) assessment team must evaluate the subject's compliance with DCD criteria set out in the Guidelines for Organ Donation after Cardiac Death in China (**Data S2. DCD Guidelines**). Subsequently, the attending physician informs donor's family that death is imminent and inevitable. Next, an organ coordinator communicates with donor's next-of-kin in accordance with Chinese laws and ethics to provide the legal, ethical and medically acceptable recommendations for implementing DCD

donation for the purposes of medical education and scientific research (**Data S3. Approval system for living organ transplantation**). In the case where the donor's next-of-kin fully understands and consent to the use of organs for donation, they would be required to sign an informed consent form. It is entirely an opt-in and voluntary procedure. Following this, the Organ Procurement Organization (OPO) team and its members, under the supervision of the donation committee personnel, proceed with organ procurement and sample collection. The study is approved by the human ethics committee of The First Affiliated Hospital of Xi'an Jiaotong University (**Data S4. Ethical Approval**), which also oversees the legal and ethical compliance of the collection process. According to the updated medical records, none of the donors previously joined with organ donor registry (as of 2023, only 0.4% of general population were registered organ donors). No reward or value were offered to families who consented for organ donation for research. The consent rate was 24.2% (33/136). This information has now been added to **Methods** section:

Materials and Methods (p.19, line 376-390):

Human subjects

All the human donors were declared dead by cardiovascular death. Major causes of death include vehicle accident, high-altitude falling, etc. (**Table S1**). Subjects were excluded if they had tumour, infectious disease, or metabolic disease. Written informed consent was obtained from each enrolled donor via next-of-kin to permit the collection and banking of samples (consent rate: 24.2% (33/136)) (**Data S1**). No organs or other specimens used in this study were obtained from executed prisoners. Sample collection was conducted under the instruction and supervision of Organ Procurement Organization of First Hospital of Xi'an Jiaotong University and the Red Cross Society of Shaanxi Province (**Data S2-3**). Samples from multiple surface organs were collected right after the harvest of liver and kidney for organ transplantation in the First Affiliated Hospital of Xi'an Jiaotong University. Characteristics of recruited 33 subjects was provided in **Table S1**. The study was approved by the Clinical Application Ethics Committee of First Affiliated Hospital of Xi'an Jiaotong University (Approval No. XJTU1AF2019LSK-059) and conducted in accordance with the Declaration of Helsinki (**Data S4**).

6. Please provide some clarification regarding the disproportionate number of males recruited for this study. Was consent lower among families of potential female donors? More demographic details regarding the circumstances of the donors' deaths would be helpful and may be pertinent in analysis of the results. For example, what was the duration of length of stay in intensive care prior to death? What was the duration between a decision to cease life sustaining interventions and commence of a DCDD pathway? Between determination of death by neurological criteria and commencement of organ procurement? What was the primary cause of death in these cases? When you note that individuals with "metabolic disease" were excluded, does this mean all those with comorbidities such as diabetes were excluded?

RESPONSE: The higher number of males recruited compared to females was because of the much lower incidence of unnatural deaths among females. At our

hospital, we receive more potential male donors that suffer accidental deaths, e.g., vehicle accidents, risk-taking activities and occupational risks (e.g., construction workers) (**Table S1**). We have also added the duration of hospitalization for all donors, and specified donors ($n=5$) who have stayed in intensive care unit (**Table S1**). Duration between the determination of cardiac death and commencement of organ procurement was strictly controlled within 75-95 min (**Table S1**). The primary causes of death are now listed in **Table S1**. All individuals with metabolic diseases, including diabetes, were excluded from our cohort.

Finally, please confirm explicitly that no organs or other specimens used in this study were obtained from executed prisoners

RESPONSE: We can confirm that no organs or other specimens used in this study were obtained from executed prisoners, and have now provided this statement in the manuscript, as follows:

Materials and Methods (p.19, line 381-382):

No organs or other specimens used in this study were obtained from executed prisoners.

Response to Reviewer #3:

The study focused on examining the microbiome in various human organs, specifically investigating the microbial communities within and between these organs. The authors collected samples from different regions of seven surface organs (oral cavity, esophagus, stomach, small intestine, appendix, large intestine, and skin) from 33 subjects, totaling 1608 samples. They used 16S rRNA sequencing techniques to analyze the microbial diversity and species and used PICRUST to infer the microbial functions. The findings of the study revealed significant variations in microbial diversity among different surface organs. They observed that certain core microbial species were consistent across different organs within the same person. The authors also delved into the microbial changes within specific regions of individual organs and identified signature microbial species, their functions, and interactions unique to each region. The study highlighted differences in the microbiomes between paired mucosa-lumen samples of the stomach, small intestine, and large intestine. Additionally, they constructed a comprehensive overview of the relationships between microbes across the digestive tract, establishing an inter-organ microbial landscape.

1. This study holds merit in its substantial sample size and the inclusion of samples from diverse body sites, which enhances its credibility. Nevertheless, a notable limitation lies in its heavy reliance on 16S data, rendering it predominantly compositional and descriptive in nature. Furthermore, a significant portion of the study's conclusions lack novelty, as they align with existing knowledge.

RESPONSE: Due to the low biomass, we were not able to characterize tissue microbiomes using metagenomic sequencing [3, 4], but applied PacBio long read HiFi sequencing to obtain 16S full-length sequences to enable species-level

analysis. Thus, our work has the following novelties not explored by previous studies:

1) We have performed dense, multisite sampling along all organs of the GI tract and skin (average of 53 sites per individual), especially the small intestine which is nearly inaccessible under conventional conditions. Our multisite collection strategy enables us to trace the inter-organ correlations in the microbial composition along different sections of GI tract.

2) We have collected paired luminal and mucosal samples at each site. Our extensive sample collection allows a more holistic view of intra-organ distribution of microbiome.

3) Besides 16S v3v4 sequencing, we have also utilized PacBio long read HiFi sequencing to obtain 16S full-length sequences in 1,030 GI samples to enable species-level analysis of the surface microbiome.

These points are now stated in **Discussion (p.14, line 279-282):**

To our knowledge, **we provided a much denser sampling of each individual along the digestive tract and skin than previous publications^{7, 11}**. Using PacBio 16S full-length sequencing data **to provide taxon information at species level**, we identified the microbiome core species with significant inter-organ relations in the human GI tract.

Discussion (p.17, line 330-331):

We collected paired mucosal and luminal samples from the GI tract. To date, most reports studied the mucosal microbiome by collecting endoscopic biopsies.

Response to Reviewer #4:

In this study, the authors focused on profiling microbiomes found in human surface organs, including the oral cavity, gastrointestinal tract (GI), and skin. They collected 1608 samples from 53 regions of the surface organs. All samples were profiled with 16S v3v4 region sequencing. Among those samples, 1030 samples from GI organs were profiled with PacBio 16S full-length HiFi sequencing in order to be analyzed at high resolution (species rank). Based on the taxonomic profiling of the samples, the authors showed 1) the difference in microbial diversity, composition, and interaction in both inter-organ and intra-organ perspectives, 2) the difference in functional pathways among different organs, and 3) microbial community differences between lumen and mucosa samples in the same organs. The experimental design and the analysis are sound and well-explained. The authors performed negative control experiments and multiple testing corrections to minimize the potential bias in their microbiome study. The findings offer a detailed view of human host-associated bacteria.

Major comments

1. In this study, the term “core microbial species” is mentioned in the abstract as species that coexisted in different organs of the same individual.

Please also formally define this concept in the main text since it is a key concept of this study.

RESPONSE: We have now formally defined “core microbial species” as microbial species that coexisted in different organs of the same individual on **page 14, line 263-264**.

2. *Given that short-read 16S sequencing of the V3-V4 region commonly does not provide sufficient taxonomic information to resolve taxonomy at the species level, the core microbial species are determined by PacBio full-length 16S sequencing, which only includes the GI organs. Is this correct?*

RESPONSE: It is correct. We have determined the core microbial species and inter-organ microbial relations based on PacBio full-length 16S sequencing of GI samples.

3. *In Figure 6, the authors state, "Areas labeled in red represent the presence of ASV on all the organs from the same individual." However, the core species were labeled in light red and dark red. Clarify the difference between light vs dark colors.*

RESPONSE: The light red color refers to those with only one type of ASV of a particular species shared among organs from the same individual, whilst the dark red color represents those with >1 types of ASVs (of that same species) shared among organs from the same individual. We have now included this information in **Figure 6** legend, as follows:

Figure 6 legend (p.35, line 710-712):

Light red color: one type of ASV of a particular species shared among organs from the same individuals. Dark red color: >1 ASVs of a particular species shared among organs from the same individuals.

4. *The authors need to formally describe the concept of upper GI and lower GI in the manuscript. The terms upper GI and lower GI were used across multiple sections, with seemingly inconsistent meanings. For example, in Figure 2B, the definition of upper GI includes duodenum(D). However, in Figure 6, duodenum(D) was included in the lower GI instead. Does including duodenum in the upper GI changes your finding?*

RESPONSE: In **Figure 2B**, the duodenum (sites DUB, MDP, and DUF) was included in the small intestine, which belongs to lower GI. Hence, the concept of upper and lower GI is consistent throughout the manuscript. We have now revised **Figure 2B** to highlight duodenum as part of the small intestine.

5. *In Figure 2C, the author provided a PCoA plot of the beta diversity of microbiomes from different organs and showed distinguishable clusters between them. However, the authors state, "In the small intestine, we observed drastic intra-organ variation in microbiome spanning between the stomach and large intestine." Since the samples are collected from different regions of the organs, and intra-organ analysis is done in Figure 4, it would be very helpful to know if colored by different regions, whether clear clusters or beta diversity shifts can be observed based on the locations of sampling across organs. More specifically, the authors should show whether the intra-organ variation in the small intestine cluster that*

spans between the stomach and large intestine clusters you observed in Figure 2C is associated with the sampling location.

RESPONSE: We have now performed the beta diversity analysis for each site of small intestine, together with the stomach and large intestine. The results have shown that intra-organ variation in the small intestine cluster spans between stomach and large intestine clusters according to the sampling location (**Figure S1D**). This data has been added to **Results**, as follows:

Results (p.8, line 141-143):

In the small intestine, we observed drastic intra-organ variation (Figure 2C), and showed that intra-organ variation in the small intestine spans between stomach and large intestine clusters according to the sampling location (Figure S1D).

6. Figure 5A shows two different analyses. In the stomach, the comparison is between all lumen and mucosa samples without specification of the sampling regions. However, in the small intestine and large intestine plots, the comparison is between selected regions where you have both lumen and mucosa samples, and regions without both lumen and mucosa samples are excluded. This analysis should be made consistent to compare apples to apples.

RESPONSE: The comparison in **Figure 5A** was performed between all lumen and mucosa samples in stomach, small intestine and large intestine, and no samples were excluded. We have marked different regions in the figure for reference only. We have now annotated the regions for gastric mucosa samples in **Figure 5A**.

7. Overall, the figure legends would benefit from additional details. Please indicate the meanings of colormaps for all heatmaps, including Figure 3B, Figure 4, Figure 6A (I'm guessing the red dot is 0.75 prevalence), and 6C (as mentioned before).

RESPONSE: We have now provided additional details in Figure legends for **Figure 3B**, **Figure 4**, and **Figure 6A**, as follows:

Figure 3 legend (p.33, line 672-674):

(B) Genus with significantly different abundance among seven organs by ALDEx2 method. **The colormap represents the average bacterial abundance.**

Figure 4 legend (p.34, line 683-684):

Selected microbes (FDR<0.05) were coloured based on their phyla.

Figure 6 legend (p.35, line 700-703; p.35, line 710-712):

(A) Bacterial prevalence in each organ by 16S full-length sequencing. ASVs with relative abundance >0.1% (~10 sequencing reads) were considered as present on the organ. **Red dot represents >50% prevalence.**

(C) **Light red colour: one type of ASV of a particular species shared among organs from the same individuals. Dark red colour: >1 ASVs of a particular species shared among organs from the same individuals.**

8. It was unclear which analyses were based on 16S full-length sequencing and which were based on 16S v3v4 sequencing. Is 16S full-length data just being used on inter-organ core species analysis (Figure 6)? For example, if that's true, since 16S full-length sequencing provides better classification accuracy and resolution,

why not use those data to compare lumen and mucosa samples (Figure 5)? Does the conclusion still hold with 16S full-length sequencing data? Please clarify.

RESPONSE: 16S full-length data were mainly used in **Figure 6**. We have now performed additional analysis in **Figure 5** using 16S full-length data (**Figure S3**). Our result showed that consistent lumen/mucosa-enriched bacteria were identified by 16S full-length data in the stomach, small intestine, and large intestine, respectively (**Figure S4**). These results are now described in **Results**, as follows: **Results (p.11, line 200-203):**

We then conducted similar analysis using 16S full-length dataset (**Figure S3**) in order to validate the above observations. We found that consistent lumen/mucosa-enriched bacteria were identified along the GI tract, including the stomach, small intestine, and large intestine (**Figure S4**).

9. *The data availability statement indicates all data is available via a genome sequence archive bioproject ID, but when accessing the link it returns". The data you are retrieving is not released in BioProject: PRJCA017513" and "Wrong share code OR this preview is Cancelled." so it seems the data are not yet made available?*

RESPONSE: The datasets generated during and/or analyzed during the current study are available in the Genome Sequence Archive (GSA) database (BioProject ID: PRJCA017513; Accession ID: HRA004781). We have now temporally released the data (>3 months, from 26/09/2023) for peer-review at <https://ngdc.cncb.ac.cn/gsa-human/s/WBV/bJx15>. Data will be permanently released after our manuscript is published.

10. *Regarding all of the source code, scripts, and outputs from the analyses performed for the study, I was unable to locate them. Similarly, no parameters were indicated for the software used (qiime, cutadapt, decontam, etc).*

RESPONSE: Source code and scripts performed for the study have now been uploaded to <https://github.com/WilsonYangLiu/DCD-bacteria.git>. The parameters of the software used were now indicated in the **Materials and Methods** section.

11. *The authors include ASV based analysis throughout, yet fail to include a discussion (or comparison to) OTU based analyses and approaches (for reference, please see: <https://www.ncbi.nlm.nih.gov/pmc/articles/PMC8870492/>). The scientific findings would be strengthened by a discussion of, or comparison to, OUT based approaches.*

RESPONSE: We have now discussed ASV- vs OTU-based analyses in **Results (p.18, line 350-353)**, as follows:

Compared to OTU-based analysis, ASV-based analysis offers advantages such as a finer resolution down to single nucleotide level²⁴. A single base difference in the 16S sequence will result in a unique ASV, thus providing a more detailed profiling of microbial diversity.

Minor comments:

1. *Figure 1A is unclear since not all samples were sequenced based on 16S full-length sequencing and requires clarification.*

RESPONSE: We have now indicated the samples for 16S v3v4 and/or 16S full-length sequencing in details in **Figure 1A**.

2. *Please add the description for the top subfigure of Figure 2C in the figure legend*

RESPONSE: We have now removed the top subfigure of **Figure 2C**, as this information was not mentioned in the manuscript.

3. *Please be more specific on Figure 3A1; what is the x-axis label? I'm guessing individual samples. How were those samples clustered?*

RESPONSE: The x-axis represents samples grouped by organs and hierarchical clustering was performed within each organ. We have revised **Figure 3A1** as suggested by *Reviewer #1*.

4. *For Figure S2, fitted curves of relative abundance are mentioned in the figure legend but not visible in the figure.*

RESPONSE: We fitted curves of relative abundance in **Figure 3A1**, but not in **Figure S2**. We have now removed this statement in **Figure S2** legend.

5. *Requires rewrite due to unclear meaning: "Moreover, we found that there are more bacteria with positive correlations between organs from the upper GI or lower GI than that of upper-to-lower GI organs (Figure 6B)" to clarify that the positive correlations are more distinguishable within upper/lower GI groups than across groups if this is indeed the case.*

RESPONSE: We have rewritten the sentence (**p.13, line 254-256**) as follows: **Moreover, bacteria with positive correlations were more distinguishable within upper or lower GI organs (e.g., esophagus and stomach: ratio=0.53; SI and LI: ratio=0.51) than between upper and lower GI organs (e.g., esophagus and LI: ratio=0.13).**

6. *Figure S3A is confusing as the coloring is based on the superclasses of pathways. However, I'm unsure if the p-values are calculated based on the superclasses or finer categories. The superclass profiles do not look very different between organs; please clarify this.*

RESPONSE: The p-values are calculated based on the superclasses by Friedman's test.

Response to Reviewer #5:

I co-reviewed this manuscript with one of the reviewers who provided the listed reports. This is part of the Nature Communications initiative to facilitate training in peer review and to provide appropriate recognition for Early Career Researchers who co-review manuscripts

RESPONSE: Noted with many thanks.

Response to Reviewer #6:

Thank you for sending me the manuscript to review, “Defining the biogeographical map and potential bacterial translocation of microbiome in human surface organs”. She et al describe a multi-organ microbiome survey focussing on 53 sites in 33 individuals, and examine the signature species and shared taxa between them as well as the inferred functional differences.

1. It is to be applauded that the study undertakes much denser sampling of each individual than is generally found in the literature. However, the authors seem to suggest (line 60, 262) that within-host translocation or microbial community changes along the gut have not been studied before, but this is not the case, for example [Vonaesch et al 2018, pubmed 30126990] specifically addresses pathological translocation between oral/stomach /duodenum/colon.

RESPONSE: We have now revised our statements as follows:

Introduction (p.4, line 60-64):

*“However, **only a few investigations deciphered the microbiome along the digestive organs with limited sampling site from the same individual**⁷ (Vonaesch, et al). To fully uncover the human microbiome, it is necessary to assess microbiome composition in surface organs of digestive system (lumen and mucosa) and skin **with much denser sampling as a whole.***

Discussion (p.14, line 279-281):

*To our knowledge, **we provided a much denser sampling of each individual along the digestive tract and skin than previous publications**^{7, 11}.*

2. There is an important aspect of the methodology that is not mentioned in the main text: I was surprised to reach page 18 before discovering that the subjects were all deceased organ donors.

RESPONSE: We have now provided this information at the beginning of the main text, as follows:

Results (p.6, line 96-99):

*We collected 1,608 samples from 7 surface organs of oral cavity (6 sites), esophagus (4 sites), stomach (5 sites), small intestine (14 sites), appendix (1 site), large intestine (13 sites) and skin (10 sites), in total comprising of 53 sites in 33 subjects (**Figure 1A**) **who were dead due to vehicle accident, high-altitude falling, etc. (Table S1).***

3. Although the authors state that as the subjects died very shortly before sampling there would not be any post mortem deterioration in the microbiome, there is no elaboration on the other associated factors that would affect the microbiome in the digestive tract and elsewhere (such as length of hospital stay, time spent in ICU, use of enteral feeding, surgical or antibiotic treatments that occurred before sampling). For example, it is known that critically ill patients in hospital commonly experience reduced gut motility, which can lead to increased bacterial translocation, a specific focus of this manuscript.

It would be helpful for the authors to address this directly in the text, and state whether the results can confidently be generalised to the microbiome of non-hospitalised living people, and if not then where the caveats lie.

RESPONSE: We have now included detailed information for all the subjects, including the length of hospital stays, time spent in ICU, and surgical or antibiotic treatments (**Table S1**). We have also performed permutational multivariate analysis of variance (PERMANOVA) to determine the effect of subject's characteristics (e.g., cause of death, length of hospitalization, ICU requirement) on the microbiome communities. We found the length of hospitalization, ICU requirement, and antibiotic treatments had significant effects on microbiome in the oral cavity, small intestine, and large intestine, but not the cause of death (**Table S3**).

Nonetheless, we believe that our data can be generalized to healthy, living subjects because 1) all donors were healthy prior to accidental death; 2) short duration of sampling after determination of death; 3) small number of subjects (n=5) requiring ICU treatment or extended hospitalization. We have now added the information to **Results** section of the revised manuscript, as follows:

Results (p.6, line 96-99):

We collected 1,608 samples from 7 surface organs of oral cavity (6 sites), esophagus (4 sites), stomach (5 sites), small intestine (14 sites), appendix (1 site), large intestine (13 sites) and skin (10 sites), in total comprising of 53 sites in 33 subjects (**Figure 1A**) who were dead due to vehicle accident, high-altitude falling, etc. (**Table S1**).

Results (p.6, line 111-116):

We next applied permutational multivariate analysis of variance (PERMANOVA) to study the effect of subject's characteristics (e.g., cause of death, length of hospitalization) on the microbiome communities. We found the length of hospitalization and antibiotic treatments had significant effects on microbiome in the oral cavity, small intestine, and large intestine, but not the cause of death (**Table S3**).

4. A note about decontamination methodology: 5% of ASVs being removed by Decontam seems quite a large proportion, higher than I would expect from exogenous contamination sources in high biomass samples. However, there is also a significant risk of inter-sample contamination (cross contamination during sample handling or PCR, as well as barcode bleed) which should not be wholesale removed.

Barcode bleed is greatly reduced by using dual-barcoding, but it is not clear from the methods whether single- or dual-indexing was performed. The readers' confidence in the Decontam removal would be improved if the authors briefly describe what was removed - for example if they're derived from exogenous sources then the removed ASVs should include environmental taxa and would be expected to show a batch effect and make up a higher proportion in the lower biomass samples.

RESPONSE: Indeed, we have used the dual-indexing method to reduce the barcode bleed. We have also provided additional details on Decontam removal

(**Table S2**), showing that the ASVs removed consist primarily of environmental taxa [5] (e.g. *Propionibacterium* (17.08%; relative abundance in negative controls), *Phyllobacterium* (6.12%), *Deinococcus* (4.87%), *Corynebacterium* (2.67%)). Consistent with your notion, Decontam removed a higher proportion of ASV in tissues samples (low biomass) than in luminal samples (high biomass) (**Figure S1B**). Moreover, levels of key contaminating ASVs in mucosal samples is on average one order of magnitude higher than that in luminal samples (**Table S2**). These findings are now described in **Results**, as follows:

Results (p.6, line 107-111):

Key contaminating ASVs consist of environmental taxa (e.g., *Propionibacterium* (17.08%; relative abundance in negative controls), *Phyllobacterium* (6.12%), and *Deinococcus* (4.87%)), and they were on average one order of magnitude higher than in mucosal samples as compared to luminal samples (**Table S2**).

5. A final minor point, but there are a number of non-standard word choices. The main example is in the title, “surface organs”, which makes sense conceptually once it is explained but does not mean what it first appears to (an external organ). Have the authors coined this phrase? If so please state that is the case or otherwise cite the original usage. It may reduce confusion in the title to add “.

RESPONSE: Surface organs are now formally described in the abstract and main text.

6. Over all the study demonstrates a dense sampling strategy that allows novel insights to be drawn regarding the compartmentalisation (or noncompartmentalisation) of these organs’ microbes. If the authors are able to address the points above I should be happy to see it published.

RESPONSE: We thank the reviewer for the positive comments.

References:

1. Yang, L., Chen, J. A comprehensive evaluation of microbial differential abundance analysis methods: current status and potential solutions. *Microbiome* **2022**, 10(1).
2. Nearing, J.T., Douglas, G.M., Hayes, M., MacDonald, J., Desai, D., Allward, N., Jones, C.M.A., Wright, R., Dhanani, A., Comeau, A.M., Langille, M.G.I. Microbiome differential abundance methods produce different results across 38 datasets. *Nature Communications* **2022**, 13(1).
3. Chiu, C.Y., Miller, S.A. *Clinical metagenomics*. *Nat Rev Genet* **2019**, 20(6), 341-355.
4. Gu, W., Miller, S., Chiu, C.Y. *Clinical Metagenomic Next-Generation Sequencing for Pathogen Detection*. *Annu Rev Pathol* **2019**, 14, 319-338.
5. Eisenhofer, R., Minich, J.J., Marotz, C., Cooper, A., Knight, R., Weyrich, L.S. Contamination in Low Microbial Biomass Microbiome Studies: Issues and Recommendations. *Trends in Microbiology* **2019**, 27(2), 105-117.

REVIEWER COMMENTS

Reviewer #1 (Remarks to the Author):

The authors have addressed most of my comments and concerns. I am concerned about the use of Friedman's test for comparing relative abundances in Figure 3A2. These are compositional data. Friedman's test is not the correct statistical test. Oddly, in Figure 3B they use ALDEX2 for differential abundance analysis.

Firstly, I do not understand the use of two different types of methods for the two data sets. Both are compositional data, why treat them differently. Secondly, why in one case are they using a method not designed for compositional data and in the other use a method designed for compositional data. Thirdly, there are several methods available in the recent literature that are better than ALDEX2. The authors should use the more recent methods.

Lastly, for correlation analysis, they should report only SECOM results because SPARCC has been demonstrated to have poor specificity and sensitivity. They could add a short sentence stating that they also considered SPARCC (data not shown) and some of the results were consistent between the two procedures.

Reviewer #2 (Remarks to the Author):

I appreciate the authors attention to my comments and the clarification of key details regarding the ethical conduct and design of the study.

Response to the comments of the referees in relation to the manuscript NCOMMS-23-23427A-Z:

COMMENTS are written in *italics* and RESPONSES in normal text.

Response to Reviewer #1:

The authors have addressed most of my comments and concerns.

1) I am concerned about the use of Friedman's test for comparing relative abundances in Figure 3A2. These are compositional data. Friedman's test is not the correct statistical test. Oddly, in Figure 3B they use ALDEX2 for differential abundance analysis. Firstly, I do not understand the use of two different types of methods for the two data sets. Both are compositional data, why treat them differently. Secondly, why in one case are they using a method not designed for compositional data and in the other use a method designed for compositional data. Thirdly, there are several methods available in the recent literature that are better than ALDEX2. The authors should use the more recent methods.

RESPONSE: We have now used ANCOM-BC2 only [Reference no. 29], a more recently developed method throughout the manuscript (**Figure 3A2** and **3B**; **Figure 4**), in place of Friedman's test and ALDEX2. The results were consistent with previous data. We have revised the **Results** section as follow:

Results (p.9, line 166-169):

We identified the signature microbes specific to each site in an organ: *Corynebacterium* and *Staphylococcus* in the extremity cluster in skin (**Figure 4A**); and *Aggregatibacter* in the jaws cluster of the oral cavity (**Figure 4B**).

2) Lastly, for correlation analysis, they should report only SECOM results because SPARCC has been demonstrated to have poor specificity and sensitivity. They could add a short sentence stating that they also considered SPARCC (data not shown) and some of the results were consistent between the two procedures.

RESPONSE: We have now replaced the SPARCC results with SECOM results as suggested (**Supplementary Fig. 7-8**) and revised the **Results** section as follow:

Results (p.12, line 220-226):

To uncover microbial interplay in each organ, we calculated pairwise microbial interactions in each organ using **SECOM method** (**Supplementary Fig. 6**). We observed that each organ has its own patterns of microbial interactions (**Supplementary Fig. 7** and **Supplementary Table 6**). Significantly different microbial interactions were observed among organs, with more co-exclusive relationships in oral cavity and large intestine, and more co-occurrent relationships in **other organs** (**Supplementary Fig. 8A**). **We also used SPARCC method, which showed consistent findings (data not shown).**

Response to Reviewer #2:

I appreciate the authors attention to my comments and the clarification of key details regarding the ethical conduct and design of the study.

RESPONSE: We thank the reviewer for the positive comments.

REVIEWERS' COMMENTS

Reviewer #1 (Remarks to the Author):

The revision addresses all my previous concerns. However, the Statistical Analysis section needs to be updated in view of the use of ANCOM-BC2 in place Friedman test and Wilcoxon rank-sum test (lines 537,538).

Response to the comments of the referees in relation to the manuscript NCOMMS-23-23427B:

COMMENTS are written in *italics* and RESPONSES in normal text.

Response to Reviewer #1:

The revision addresses all my previous concerns. However, the Statistical Analysis section needs to be updated in view of the use of ANCOMBC2 in place Friedman test and Wilcoxon rank-sum test (lines 537,538).

RESPONSE: We thank the reviewer for the positive comment. We have now updated the **Statistical Analysis** section with the use of ANCOMBC2 to replace Friedman test. The Wilcoxon rank-sum test was applied for differential testing of α -diversity. We have revised the **Statistical Analysis** section as follow:

Statistical Analysis (p.27, line 494):

Statistical significance tests, including Wilcoxon signed-rank test, ANOVA permutation test, **ANCOM-BC2**, SECOM, and SparCC correlation test, were performed using open-source R software.